# METAPHYSICA: CAUSALITY-AWARE ROBUSTNESS TO OOD INITIAL CONDITIONS IN PHYSICS-INFORMED MACHINE LEARNING

## ABSTRACT

A fundamental challenge in physics-informed machine learning (PIML) is the design of robust PIML methods for out-of-distribution (OOD) forecasting tasks, where the tasks require learning-to-learn from observations of the same (ODE) dynamical system with different unknown parameters, and demand accurate forecasts even under initial conditions outside the training support. In this work we propose a solution for such tasks, which we define as a meta-learning procedure for causal structural discovery (including invariant risk minimization). Using three different OOD tasks, we empirically observe that the proposed approach significantly outperforms existing state-of-the-art PIML and deep learning methods.

## 1 INTRODUCTION

Physics-informed machine learning (PIML) (e.g., (Willard et al., 2020; Xingjian et al., 2015; Lusch et al., 2018; Yeo & Melnyk, 2019; Raissi et al., 2018; Kochkov et al., 2021)) seeks to combine the strengths of physics and machine learning models and has positively impacted fields as diverse as biological sciences (Yazdani et al., 2020), climate science (Faghmous & Kumar, 2014), turbulence modeling (Ling et al., 2016; Wang et al., 2020a), among others. PIML achieves substantial success in tasks where the test data comes from the same distribution as the training data (*in-distribution tasks*).

Unlike the PIML work described above, this paper considers an out-of-distribution (OOD) change in the initial system state of the dynamical system, possibly with different train and test distribution supports (illustrated in Figure 1(a,b)). In this setting, we observe that existing state-of-the-art PIML models perform significantly worse than their performance in-distribution, even in PIML methods designed with OOD robustness in mind (Wang et al., 2021b; Kirchmeyer et al., 2022). This is because the standard ML part of PIML, which tends to learn spurious associations, will perform poorly in our OOD setting. We then propose a promising solution: Combine *meta learning* with *causal structure discovery* to learn an ODE model that is robust to OOD initial conditions. In our OOD tasks, OOD robustness means that the robustness is tied to interventions over the initial conditions of the system, not on arbitrary interventions as the system evolves from the initial state. This is an important distinction. There can be multiple ODE models that will be equally OOD robust, and robust ODEs may not correctly predict system trajectories under arbitrary system interventions besides the initial state (Rubenstein et al. (2016) discusses the effect of arbitrary interventions in physics models).

**Contributions** This work proposes a *hybrid transductive-inductive modeling approach* learning for more robust ODEs using *meta learning* and *causal structure discovery* (e.g., via $L_1$ regularization (Zheng et al., 2018), which can be combined with invariant risk minimization (Arjovsky et al., 2019; Krueger et al., 2021)). More precisely, our contributions are:

1. We show that state-of-the-art PIML and deep learning methods fail in test examples with OOD initial conditions. Prior work (Wang et al., 2021a) showed that deep learning-only methods fail in OOD tasks, and argued physics models and PIML methods would succeed, including a proposed OOD solution (Wang et al., 2021b). Here we show that PIML methods also fail (or perform poorly) OOD, including the solution in Wang et al. (2021b).

2. *We proposed a hybrid transductive-inductive learning framework for ODEs via meta learning*: As in transductive methods, we will consider each training and test examples as separate tasks, but

like inductive methods, the tasks are dependent and knowledge can be transferred between the learned ODEs. By *meta learning* we mean the definition in (Thrun & Pratt, 1998, Chapter 1.2), where given: (a) a family of $M$ tasks (a task is a single experiment in our setting), $i = 1, \ldots, M$; (b) training experience for each task $i \in \{1, \ldots, M\}$, which for us are the time series observations of an experiment $\mathbf{X}_{t_0}^{(i)}, \ldots, \mathbf{X}_{t_T}^{(i)}$, and; (c) a family of performance measures (e.g., one for each task) described by the risk function $R^{(i)}$; our algorithm will *meta learn* such that performance at each task improves with experience (more observations) and with the number of tasks (number of experiments). For an algorithm to fit this definition, there must be a transfer of knowledge between multiple tasks that has a positive impact on expected task performance across all tasks.

3. *Learning ODEs as structural causal discovery.* In order to learn an ODE that is robust OOD changes in initial conditions (with possibly non-overlapping training and test distribution supports), we define a family of structural causal models and perform a structural causal search in order to find the correct model for our task (which is assumed to be in the family). We test common structural causal discovery approaches for linear models: $\ell_1$-regularization with and without an invariant risk minimization-type objective, which we observe achieve similar empirical results.

The proposed method is then empirically validated using three commonly-used simulated physics tasks (with measurement noise): Damped pendulum systems (Yin et al., 2021), predator-prey systems (Wang et al., 2021a), and epidemic modeling (Wang et al., 2021a), all under both constant ODE parameters and varying ODE parameters per experiment. ODE parameters between train and test experiments have the same distribution but non-overlapping (OOD) initial condition distributions.

## 2 DYNAMICAL SYSTEM FORECASTING AS A META LEARNING TASK

In this section we formally describe the task of forecasting a dynamical system with a focus on the out-of-distribution initial condition scenario.

**Definition 1 (Dynamical system forecasting task)** *In what follows we describe our task:*

1. **Training data (depicted in Figure 1(a)):** *In training, we are given a set of $M$ experiments, which we will denote as $M$ tasks. Task $i \in \{1, \ldots, M\}$ has an associated (hidden) environment $e^{(i)}$. Different tasks can have the same environment. Let $\mathcal{T}^{(i)} := \mathbf{X}_{t_0}^{(i)}, \ldots, \mathbf{X}_{t_{T^{(i)}}}^{(i)}$ denote the noisy observations of our dynamical system, with $\mathbf{X}_t^{(i)} := \boldsymbol{x}_t^{(i)} + \boldsymbol{\varepsilon}_t^{(i)}$, where*

$$\frac{d\boldsymbol{x}_t^{(i)}}{dt} = \psi(\boldsymbol{x}_t^{(i)}; \boldsymbol{W}^{(i)*}, \boldsymbol{\xi}^*) \,, \tag{1}$$

*$\{t_0, \ldots, t_{T^{(i)}}\}$ are regularly-spaced discrete time steps, $\boldsymbol{x}_t^{(i)} \in \mathbb{R}^d$ is the (hidden) state of the system at time $t$ during experiment (task) $i$, $\boldsymbol{\varepsilon}_t^{(i)}$ are independent zero-mean Gaussian noises, $\psi$ is an unknown deterministic function with hidden ground truth parameters $\boldsymbol{W}^{(i)*} \sim P(\boldsymbol{W}^*)$ and $\boldsymbol{\xi}^*$, where the global task-independent parameters $\boldsymbol{W}^*$ and $\boldsymbol{\xi}^*$ are also hidden. Regularly spaced intervals are not strictly necessary for our method, but it makes its implementation simpler. **Initial conditions:** The distribution of initial conditions $\mathbf{X}_{t_0}^{(i)} \sim P(\mathbf{X}_{t_0} \mid E = e^{(i)})$ of task $i$ may depend on its environment. The unknown parameters $\boldsymbol{\xi}^*$ remain constant across environments.*

2. **Test data ((depicted in Figure 1(b)):** *At test, we are given an observed initial sequence $\widetilde{\mathcal{T}}^{(M+1)} := \mathbf{X}_{t_0}^{(M+1)}, \ldots, \mathbf{X}_{t_r}^{(M+1)}$, where $r$ is generally small, of the dynamical system*

$$\frac{d\boldsymbol{x}_t^{(M+1)}}{dt} = \psi(\boldsymbol{x}_t^{(M+1)}; \boldsymbol{W}^{(M+1)*}, \boldsymbol{\xi}^*)$$

*with initial condition $\mathbf{X}_{t_0}^{(M+1)} \sim P(\mathbf{X}_{t_0} \mid E = e^{(M+1)})$ and (unknown) system parameters $\boldsymbol{W}^{(M+1)*} \sim P(\boldsymbol{W}^*)$ and hidden global parameters $\boldsymbol{\xi}^*$ the same as in training. **Our task is to predict $\mathbf{X}_{t_{r+1}}^{(M+1)}, \ldots, \mathbf{X}_{t_{T^{(M+1)}}}^{(M+1)}$ from the initial observations $\widetilde{\mathcal{T}}^{(M+1)}$, using the inductive knowledge obtained from the training data.***

*3.* ***Out-of-distribution initial conditions:*** *Initial conditions in training $\{P(\mathbf{X}_{t_0} \mid E = e^{(i)})\}_{i=1}^M$, can be different from initial conditions in test $P(\mathbf{X}_{t_0} \mid E = e^{(M+1)})$ with possibly non-overlapping support due to the presence of an unseen environment in training.*

In training, we are given trajectories that may have (a) different initial conditions, and (b) different unknown ODE system parameters. We observe a test trajectory (indexed by $M + 1$) from time $t = t_0, \ldots, t_r$ and we wish to forecast its future after time $t_r$. The test trajectory can have an OOD initial condition but in-distribution ODE parameter $W^{(M+1)*}$ with an unknown value.

**Illustrative example.** Figure 2a shows an example of an out-of-distribution task for forecasting the motion of a pendulum with friction. **(1.)** The state $\mathbf{X}_t = [\theta_t, \omega_t] \in \mathbb{R}^2$ describes the angle made by the pendulum with the vertical and the corresponding angular velocity at time $t$. The true (unknown) function $\psi$ describing this dynamical system is given by $\psi([\theta_t, \omega_t]; \boldsymbol{W}^*) = [\omega_t, -\alpha^{*2} \sin(\theta_t) - \rho^* \omega_t]$ with $\boldsymbol{W}^* = (\alpha^*, \rho^*)$ denoting the parameters relating to the pendulum's period and the damping coefficient. **(2.)** In training, we observe $M$ (noisy) trajectories of motion over discrete time steps $t = 0, 0.1, \ldots, 10$ from experiments (tasks) where a pendulum is dropped with no angular velocity. **(3.)** In training, each experiment is performed by dropping different pendulums (i.e., $\boldsymbol{W}^{(i)*} \sim P(\boldsymbol{W}^*)$) from angles $0 < \theta_{t_0} < \pi/2$. **(4.)** In test, the experiment is repeated with a different distribution over the initial dropping angles, $\pi - 0.1 < \theta_{t_0} < \pi$ (nearly vertical angles) with small angular velocities. The test trajectory is observed over a smaller time window $t = 0, 0.1, \ldots, 3.3$ and the forecasting task is to predict the future states of the pendulum till time $t = 10$.

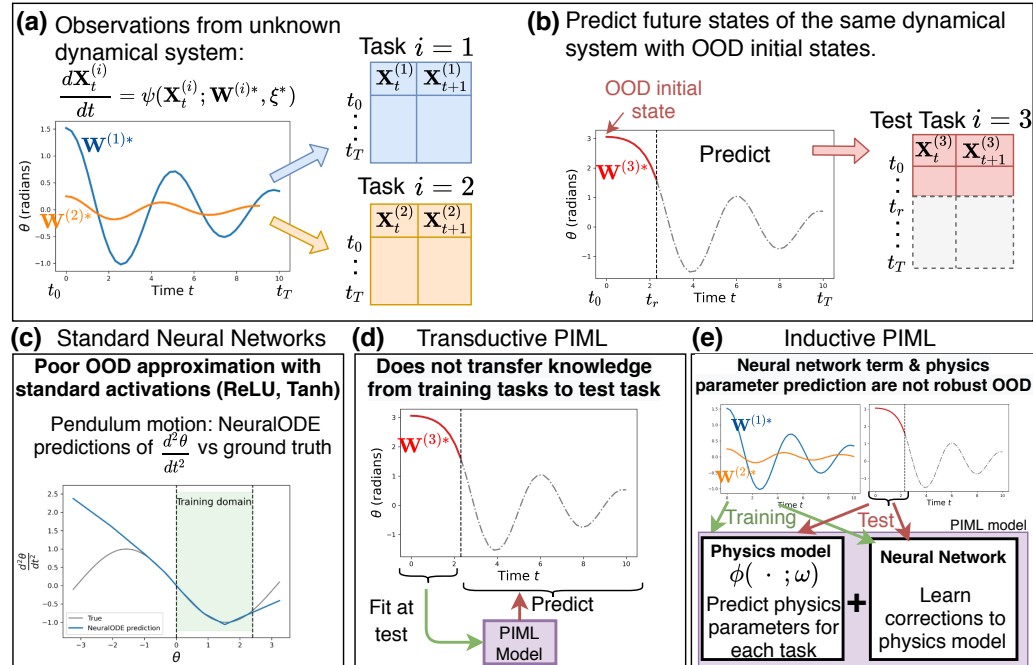

Figure 1: Dynamical system OOD problem definition and traditional approaches to address it. **(a)** Training data consists of multiple observations from the same dynamical system with different parameters $\boldsymbol{W}^{(i)*}$. Each training curve can be seen as a different task $i$ where the goal is to predict $\mathbf{X}_{t+1}^{(i)}$ from $\mathbf{X}_t^{(i)}$ for all $t$. **(b)** At test, we are given observations till $t_r$ (red solid) and the goal is to predict the future observations till $t_T$ (gray dashed). **(c)** Shows OOD failure of a standard neural network (NeuralODE (Chen et al., 2018)) for dynamical system forecasting. When trained to predict the motion of damped pendulum, the model predicts accurately in the training domain (green shaded), but predicts a linear function outside the training domain. **(d)** Transductive PIML methods (e.g., (Raissi et al., 2017a; Brunton et al., 2016)) are not able to transfer knowledge from training tasks to a test task with different $\boldsymbol{W}^*$. Thus, these models can be fit only using test observations till time $t_r$ ignoring the training data. **(e)** Inductive PIML methods (e.g., (Yin et al., 2021; Mehta et al., 2021)) use a known (possibly incomplete) physics model $\phi(\ \cdot \ ; \omega)$ and inductively predict its parameters $\omega$ for each task, typically using a neural network. However, predicting these physics parameters at test this way is not robust. Furthermore, they use a neural network term to correct for the incomplete physics model and face the same robustness issue discussed in **(c)**.

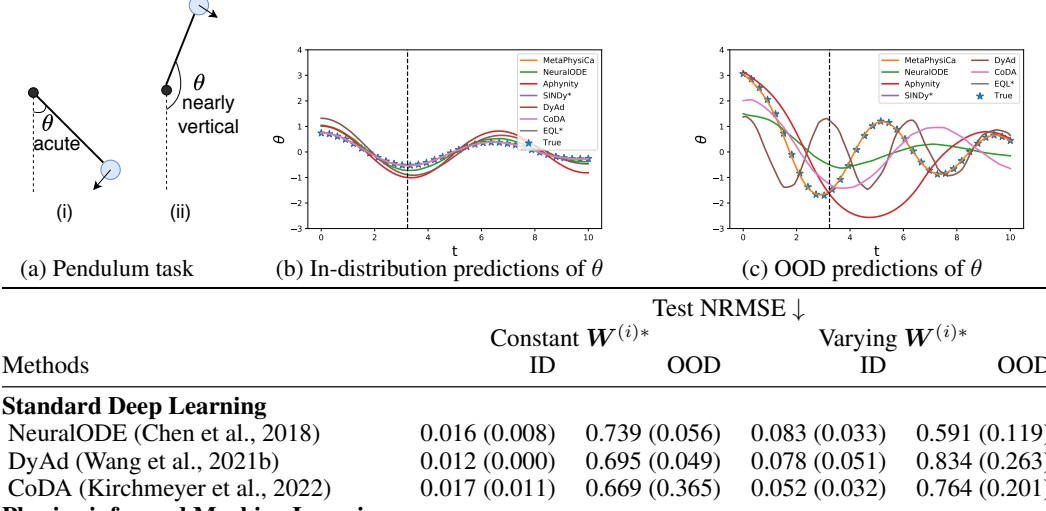

(a) Pendulum task     (b) In-distribution predictions of $\theta$     (c) OOD predictions of $\theta$

| | Test NRMSE ↓ | | | |
| --- | --- | --- | --- | --- |
| | Constant $\boldsymbol{W}^{(i)*}$ | | Varying $\boldsymbol{W}^{(i)*}$ | |
| Methods | ID | OOD | ID | OOD |
| **Standard Deep Learning** | | | | |
| NeuralODE (Chen et al., 2018) | 0.016 (0.008) | 0.739 (0.056) | 0.083 (0.033) | 0.591 (0.119) |
| DyAd (Wang et al., 2021b) | 0.012 (0.000) | 0.695 (0.049) | 0.078 (0.051) | 0.834 (0.263) |
| CoDA (Kirchmeyer et al., 2022) | 0.017 (0.011) | 0.669 (0.365) | 0.052 (0.032) | 0.764 (0.201) |
| **Physics-informed Machine Learning** | | | | |
| APHYNITY (Yin et al., 2021) | 0.047 (0.022) | 0.803 (0.168) | 0.097 (0.020) | 0.970 (0.384) |
| SINDy (Brunton et al., 2016) | NaN* | NaN* | NaN* | NaN* |
| EQL (Martius & Lampert, 2016) | NaN* | NaN* | NaN* | NaN* |
| MetaPhysiCa **(ours)** | 0.028 (0.005) | **0.078 (0.090)** | 0.049 (0.002) | **0.070 (0.011)** |

(d) Normalized RMSE ↓ of test predictions from different methods in two cases: with $\boldsymbol{W}^{(i)*}$ constant or varying across environments $i$. NaN* indicates that the model returned errors during test-time predictions, for example, because the learnt ODE was too stiff (numerically unstable) to solve.

Figure 2: **(a)** Predict pendulum motion from noisy observations: (i) in-distribution, when dropped from acute angles and (ii) out-of-distribution, when dropped from nearly vertical angles. **(b, c)** shows example ground truth curves (blue stars) in- and out-of-distribution along with predictions from different models. While most tested methods perform well in-distribution, only MetaPhysiCa (orange) closely follows the true curve OOD and all other methods are terribly non-robust. **(d)** Standard deep learning methods and physics-informed machine learning methods fail to forecast accurately out-of-distribution. On the other hand, **MetaPhysiCa outputs up to** $8.5\times$ **more robust OOD predictions.**

## 3 RELATED WORK & CHALLENGES WITH EXISTING APPROACHES

Next we describe different classes of existing approaches that are commonly used for the dynamical system forecasting (Definition 1) and their inherent challenges out-of-distribution.

### 3.1 STANDARD NEURAL NETWORKS METHODS

Deep learning's ability to model complex phenomena has allowed it to make great strides in a number of physics applications (Lusch et al., 2018; Yeo & Melnyk, 2019; Kochkov et al., 2021; Dang et al., 2022; Brandstetter et al., 2022b). However, standard deep learning methods are known to learn spurious correlations and tend to fail when the test distribution of the inputs are different from that observed in training (Wang et al., 2021a; Geirhos et al., 2020). Figure 2 depicts the out-of-distribution failure of several deep learning methods from NeuralODE (Chen et al., 2018) to more complex meta learning approaches (Wang et al., 2021b; Kirchmeyer et al., 2022) in our running damped pendulum example (more details of the experiment is in Section 5).

In standard deep learning tasks, Xu et al. (2021) show that an MLP's failure to extrapolate to out-of-distribution can be traced to an absence of algorithmic alignment, which is an appropriate combination of basis and activation functions within the architecture for the task. For example, the outputs of an MLP with ReLU activations will be linear far from the training domain even when trained to predict a sine/quadratic function. For dynamical system forecasting, our Figure 1(c) depicts the results of a similar experiment for a standard sequence model (NeuralODE): the model can approximate the target sine function in the training domain (green region) but predicts a linear function far outside the training domain. This means that *PIML also needs algorithmic alignment (i.e., to include appropriate basis functions) in order to make accurate forecasts in OOD tasks.*

### 3.2 PHYSICS-INFORMED MACHINE LEARNING (PIML) METHODS

To alleviate the challenges described above for standard neural networks, several physics-informed machine learning (PIML) methods have been proposed (e.g., (Willard et al., 2020; Wang et al., 2020a; Faghmous & Kumar, 2014; Daw et al., 2017)) that utilize physics-based domain knowledge about the dynamical system in consideration for better predictions. The type of physics-based knowledge vary

across methods, for example, **(a)** a dictionary of basis functions (e.g., $\sin$, $\cos$, $\frac{d}{dt}$) (Schmidt & Lipson, 2009; Brunton et al., 2016; Martius & Lampert, 2016; Raissi, 2018; Cranmer et al., 2020a) related to the task, **(b)** a completely specified physics model (Raissi et al., 2017a; Raissi, 2018; Jiang et al., 2019) or with missing terms (Yin et al., 2021), and **(c)** different domain-specific physical constraints such as energy conservation (Greydanus et al., 2019; Cranmer et al., 2020b), symmetries (Wang et al., 2020b; Finzi et al., 2021; Brandstetter et al., 2022a). While these PIML methods improve upon standard neural networks, Figure 2 shows that they are generally not designed for OOD forecasting tasks. To precisely study the reasons for this failure, we categorize these methods into inductive and transductive methods based on requirements over the dynamical system parameters $\boldsymbol{W}^*$.

*Transductive PIML methods.* Transductive inference focuses on predicting missing parts from the training data. In PIML, transductive inference methods treat *each training and test examples* as unrelated tasks, hence OOD generalization tends to be less of a challenge in transductive methods. For instance, SINDy (Brunton et al., 2016), EQL (Martius & Lampert, 2016), and related methods (Raissi, 2018; Chen, 2021), learn the ODE equation based on a dictionary of basis functions for a specific parameter $\boldsymbol{W}^{(i)*}$. These transductive methods, however, do not transfer knowledge learnt in training to predicting test examples with a different in-distribution $\boldsymbol{W}^{(j)*}$. This forces these methods to forecast simply based on the initial observations of the test task alone, often leading to poor performance. Figure 1(d) illustrates this case where a transductive method (unsuccessfully) tries to learn the unknown parameter $\boldsymbol{W}^{(3)*}$ of the test task from a few initial test observations. Another class of transductive methods (Raissi et al., 2017a;b; Yu et al., 2022) assume all the physics parameters $\boldsymbol{W}^*$ of all experiments to remain constant across all training and test tasks, regularizing neural networks to respect a given physics model. They have been shown to be challenging to train for harder differential equations (Krishnapriyan et al., 2021) or return trivial solutions (Leiteritz & Pflüger, 2021). Recently, Causal PINNs (Wang et al., 2022) mitigate some of these training challenges by ensuring that, for any time $t$, predictions at time less than $t$ are accurately resolved before predictions at time $t$. Not only will these methods perform poorly in-distribution if different experiments have different physics parameters, they also do not allow for causal interventions to variables in the dynamical system.

*Inductive PIML.* Taking the opposite approach, *inductive inference* focuses on learning rules from the training data that can be applied to unseen test examples. Inductive methods dominate PIML approaches but are fragile OOD, since the learned rules are learned within the scope of the training data and are not guarantee to work outside the training data scope. For example, APHYNITY (Yin et al., 2021) and NDS (Mehta et al., 2021) are such inductive methods that augment a neural network to a known incomplete physics model where the parameters of the physics model are predicted inductively using a recurrent network. As illustrated in Figure 1(e), these methods are able to learn from training tasks with different true parameters $\boldsymbol{W}^{(i)*}$. However, in our experiments, APHYNITY often returns incorrect physics parameters OOD (see Figure 2c). Further, the augmented neural network suffers from the same issues discussed in Section 3.1 leading to poor OOD performance as seen in Figure 2.

With these key reasons identified for the fragility of existing methods to OOD initial conditions, next we propose an approach (*MetaPhysiCa*) that is more robust to these challenges and outputs more robust predictions out-of-distribution, while also giving accurate predictions in-distribution.

# 4 PROPOSED APPROACH: METAPHYSICA

In what follows we describe *MetaPhysiCa*, our proposed approach. We start with the description of a family of causal models, then explain how meta learning allows us to perform a hybrid transductive-inductive approach for improved OOD accuracy.

## 4.1 STRUCTURAL CAUSAL MODEL

We describe the dynamical system using a deterministic structural causal model (Peters et al., 2022) with measurement noise over the observed states and explicitly define the assumptions over the unknown function $\psi$ in Definition 1.

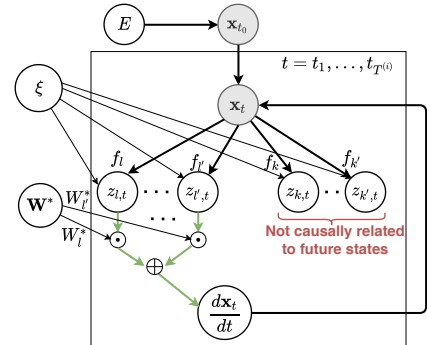

Figure 3: Deterministic SCM for a dynamical system with measurement noise. The dynamics is defined via an unknown linear combination of basis functions.

The causal diagram is depicted in Figure 3 in the plated notation iterating over time $t = t_0, \ldots, t_{T^{(i)}}$ for each task $\mathcal{T}^{(i)}$. As before, the state of the dynamical system is $\mathbf{X}_t^{(i)} \in \mathbb{R}^d$ for task $i$. We note that our SCM may not necessarily be the true SCM, but rather a SCM that is indistinguishable from the true one w.r.t. interventions limited to changes in the environment variable $E$ that affects the initial conditions $\mathbf{X}_{t_0}^{(i)}$. We define the causal process at each time step $t$ for $i$-th task as follows. Let $f_k(\cdot; \boldsymbol{\xi}_k) : \mathbb{R}^d \to \mathbb{R}, 1 \leq k \leq m$, be $m$ linearly independent basis functions each with a separate set of parameters $\boldsymbol{\xi}_k^*$ acting on an input state $\boldsymbol{x}_t^{(i)}$. Examples of such basis functions include trigonometric functions like $f_1(\boldsymbol{x}_t^{(i)}; \boldsymbol{\xi}_1^*) = \sin(\xi_{1,1} x_{t,1}^{(i)} + \xi_{1,2})$, polynomial functions like $f_2(\boldsymbol{x}_t^{(i)}; \boldsymbol{\xi}_2) = x_{t,1}^{(i)} x_{t,2}^{(i)}$, and so on. The corresponding outputs from these basis are shown as $z_{k,t}^{(i)} := f_k(\boldsymbol{x}_t^{(i)}; \boldsymbol{\xi}_k)$ in Figure 3. The derivative $d x_{t,j}^{(i)}/dt$ for a particular dimension $j \in \{1, \ldots, d\}$ is only affected by a few (unknown) basis function outputs $z_{k,t}^{(i)}$ (green arrows in Figure 3) and is a linear combination of these selected basis functions with coefficients $\boldsymbol{W}^{(i)*}$. However, these selected basis functions and their corresponding parameters $\boldsymbol{\xi}$ are assumed to be invariant across all the tasks, i.e., $d x_{t,j}^{(i)}/dt, j \in \{1, \ldots, d\}$, is defined using the same basis functions for all $i = 1, \ldots, M$. Finally, the derivatives dictate the next state of the dynamical system. We observe the dynamical system with independent additive measurement noise $\mathbf{X}_t^{(i)} := \boldsymbol{x}_t^{(i)} + \boldsymbol{\varepsilon}_t^{(i)}$, where $\boldsymbol{\varepsilon}_t^{(i)} \sim \mathcal{N}(\mathbf{0}, \sigma_\varepsilon^2 I)$.

We assume that we are given the collection of $m$ possible basis functions $f_k(\cdot; \boldsymbol{\xi}), k = 1, \ldots, m$, $m \geq 2$, with unknown $\boldsymbol{\xi}$ and *no prior knowledge of which* $\{f_k\}_{k=1}^m$ *causally influence* $d x_t^{(i)}/dt$. The need for basis functions stems from extensive experimentation and our analysis in Section 3.1, where we show that appropriate basis functions must be incorporated within the architecture in order to extrapolate to OOD scenarios (see Figure 1(c)).

## 4.2 META LEARNING & MODEL ARCHITECTURE

Given the training data $\{(\boldsymbol{x}_t^{(i)})_t\}_{i=1}^M$ generated from the unknown SCM described above, our goal is three-fold: **(a)** discover the true underlying causal structure, i.e., which of the edges $z_{k,t} \to d x_{t,j}/dt$ exist for $j = 1, \ldots, d$, **(b)** learn the global parameters $\boldsymbol{\xi}$ that parameterize the relevant basis functions, and **(c)** learn the task-specific parameters $\boldsymbol{W}^{(i)*}$ that act as coefficients in linear combination of the selected basis functions. In the following, we propose a meta-learning framework that introduces structure (gate) parameters $\Phi$ that are shared across tasks and task-specific coefficients $\boldsymbol{W}^{(i)}$ that vary across the tasks

$$\frac{d\hat{\mathbf{X}}_t^{(i)}}{dt} = (\boldsymbol{W}^{(i)} \odot \Phi) F(\hat{\mathbf{X}}_t^{(i)}; \boldsymbol{\xi}) , \tag{2}$$

where $\odot$ is the Hadamard product and

- $F(\hat{\mathbf{X}}_t^{(i)}; \boldsymbol{\xi}) := \begin{bmatrix} f_1(\hat{\mathbf{X}}_t^{(i)}; \boldsymbol{\xi}_1) & \cdots & f_m(\hat{\mathbf{X}}_t^{(i)}; \boldsymbol{\xi}_m) \end{bmatrix}^T$ is the vector of outputs from the basis functions with parameters $\boldsymbol{\xi}$,
- $\Phi \in \{0, 1\}^{d \times m}$ are the learnable parameters governing the global causal structure across all tasks such that $\Phi_{j,k} = 1$ iff edge $z_{k,t} \to d x_{t,j}/dt$ exists in Figure 3,
- $\boldsymbol{W}^{(i)} \in \mathbb{R}^{d \times m}$ are task-specific parameters that act as coefficients in linear combination of the selected basis functions.

Next we describe a procedure to obtain the structure parameters $\Phi$. Finding whether an edge exists or not in the causal graph is known as the causal structure discovery problem (e.g., Heinze-Deml et al. (2018)). We use a score-based causal discovery approach (e.g., Huang et al. (2018)) where we assign a score to each possible causal graph. We wish to find the *minimal* causal structure, i.e., with the least number of edges, that also fits the training data. This balances the complexity of the causal structure with training likelihood, and avoids overfitting the training data. A sparse structure for $\Phi$ implies fewer terms in the RHS of the learnt equation for the derivatives in Equation (2). Several causal discovery approaches have been proposed that learn such minimal causal structure via continuous optimization (Zheng et al., 2018; Ng et al., 2022). We use the log-likelihood of the training data with $\ell_1$-regularization term to induce sparsity that is known to perform well for general causal structure discovery tasks (Zheng et al., 2018). Note that since the direction of all the edges are known (i.e., $z_{k,t} \to d x_{t,j}/dt$), we do not need the acyclicity constraints and the causal graph is uniquely identified by its Markov equivalence class (Pearl, 2009, Chapter 2).

The prediction error is given by $R^{(i)}(\boldsymbol{W}^{(i)}, \Phi, \boldsymbol{\xi}) := \frac{1}{T^{(i)}+1} \sum_{t=t_0}^{t_{T^{(i)}}} ||\hat{\mathbf{X}}_t^{(i)} - \mathbf{X}_t^{(i)}||_2^2$ where $\hat{\mathbf{X}}_t^{(i)} = \mathbf{X}_{t_0}^{(i)} + \int_{t_0}^t (\boldsymbol{W}^{(i)} \odot \Phi) F(\hat{\mathbf{X}}_\tau^{(i)}; \boldsymbol{\xi}) d\tau$ are the predictions obtained using an ODE solver to integrate Equation (2). In practice however, we found the squared loss directly between the predicted and estimated ground truth derivatives, i.e., $\widetilde{R}^{(i)}(\boldsymbol{W}^{(i)}, \Phi, \boldsymbol{\xi}) = \frac{1}{T^{(i)}+1} \sum_{t=t_0}^{t_{T^{(i)}}} ||d\hat{\mathbf{X}}_t^{(i)}/dt - d\mathbf{X}^{(i)}/dt||_2^2$, leads to a stable learning procedure with better accuracy in-distribution and OOD. As discussed before, we use an $\ell_1$-regularization term $||\Phi||_1$ to learn a causal structure with the fewest possible edges $z_{k,t} \to d\boldsymbol{x}_{t,j}/dt, j = 1, \ldots, d$, while minimizing the prediction error in training. We also use $\ell_1$-regularization on the task-specific parameters, $||\boldsymbol{W}^{(i)}||_1$, to learn a simpler model within each task $i$, if possible, than the one learnt globally for all tasks via $\Phi$.

Our structure discovery task comes with an additional challenge as the training tasks could have been obtained under different (hidden) environments (as defined in Definition 1). While there are score-based (discrete optimization) approaches (Ghassami et al., 2018; Perry et al., 2022) for such non-IID data, aforementioned approaches based on continuous optimization (e.g., (Zheng et al., 2018)) are not guaranteed to learn the correct structure. For example, they may output a structure that is optimal for one environment consisting of a large number of training tasks but suboptimal for other environments. Our goal then is to learn a structure that minimizes the prediction error across all environments simultaneously, similar to learning robust representations via invariant risk minimization-type methods (Arjovsky et al., 2019; Krueger et al., 2021). Since the environment $e^{(i)}$ of a particular task $i$ is hidden to our approach, we use a modified V-REx regularization (Krueger et al., 2021) that minimizes the variance of prediction errors across tasks instead of environments, focusing on robustness to the worst-case scenario (that all tasks have unique environments).

Now we are ready to describe our final optimization objective. Similar to standard meta-learning objectives (Finn et al., 2017; Franceschi et al., 2018; Hospedales et al., 2021), we propose a bi-level objective that optimizes the structure parameters $\Phi$ and the global parameters $\xi$ in the outer-level, and the task-specific parameters $\boldsymbol{W}^{(i)}$ in the inner-level as follows

$$\hat{\Phi}, \hat{\boldsymbol{\xi}} = \underset{\Phi, \boldsymbol{\xi}}{\arg\min} \frac{1}{M} \sum_{i=1}^M R^{(i)}(\hat{\boldsymbol{W}}^{(i)}, \Phi, \boldsymbol{\xi}) + \lambda_\Phi ||\Phi||_1 + \lambda_{\text{REx}} \text{Variance}(\{R^{(i)}(\hat{\boldsymbol{W}}^{(i)}, \Phi, \boldsymbol{\xi})\}_{i=1}^M)$$

$$\text{s.t. } \hat{\boldsymbol{W}}^{(i)} = \underset{\boldsymbol{W}^{(i)}}{\arg\min} R^{(i)}(\boldsymbol{W}^{(i)}, \Phi, \boldsymbol{\xi}) + \lambda_{\boldsymbol{W}} ||\boldsymbol{W}^{(i)}||_1, \quad \forall i = 1, \ldots, M, \quad (3)$$

where $\lambda_\Phi$, $\lambda_{\boldsymbol{W}}$ and $\lambda_{\text{REx}}$ are hyperparameters. While the exact bi-level optimization in Equation (3) is challenging to solve due to the lack of closed-form solution for the inner optimization, it can be approximated by alternate SGD steps for $(\Phi, \boldsymbol{\xi})$ and $\{\boldsymbol{W}^{(i)}\}_{i=1}^M$ in outer and inner loops respectively (Borkar, 1997; Chen et al., 2021). In our experiments, jointly optimizing $\Phi, \boldsymbol{\xi}$ and $\boldsymbol{W}^{(i)}, i = 1, \ldots, M$, instead resulted in comparable performance with considerable computational benefits over alternating SGD. The discrete structure parameters $\Phi$ can be approximated using (stochastic) Gumbel-Softmax variables (Jang et al., 2017; Ng et al., 2022) or using deterministic binarization techniques (Courbariaux et al., 2015; 2016). We use the latter and reparameterize $\Phi_{j,k} := \mathbf{1}(\sigma(\widetilde{\Phi}_{j,k}) > 0.5)$ where $\Phi' \in \mathbb{R}^{d \times m}$, $\sigma(\cdot)$ is the sigmoid function, and the gradients are estimated via a straight-through-estimator.

*Hyperparameter selection:* We choose the hyperparameters $\lambda_\Phi, \lambda_{\boldsymbol{W}}, \lambda_{\text{REx}}$ that result in sparsest model (i.e., with the least $||\hat{\Phi}||_0$) while achieving validation loss within 5% of the best validation loss in held-out *in-distribution* validation data. The use of *in-distribution data for validation* is key requirement since in OOD tasks one *does not* have access to samples from the test distribution. Additional implementation details are provided in Appendix B.

### 4.3 TRANSDUCTIVE TEST-TIME ADAPTATION WITH INDUCTIVE REGULARIZATION

Finally, given a test task $\widetilde{\mathcal{T}}^{(M+1)} = (\mathbf{X}_{t_0}^{(M+1)}, \ldots, \mathbf{X}_{t_r}^{(M+1)})$ with the unknown ground-truth parameters $\boldsymbol{W}^{(M+1)*} \sim P(\boldsymbol{W}^*)$ as defined in Definition 1, we adapt the learnt model's task-specific parameters $\boldsymbol{W}^{(M+1)}$ by optimizing the following while keeping $\hat{\Phi}, \hat{\boldsymbol{\xi}}$ fixed

$$\hat{\boldsymbol{W}}^{(M+1)} = \underset{\boldsymbol{W}^{(M+1)}}{\arg\min} \frac{1}{t_r + 1} \sum_{t=t_0}^{t_r} ||\hat{\mathbf{X}}_t^{(M+1)} - \mathbf{X}_t^{(M+1)}||_2^2 + \lambda_{\boldsymbol{W}} ||\boldsymbol{W}^{(M+1)}||_1 \quad (4)$$

| | Test Normalized RMSE (NRMSE) ↓ | | | |
| --- | --- | --- | --- | --- |
| | Constant $\boldsymbol{W}^{(i)*}$ | | Varying $\boldsymbol{W}^{(i)*}$ | |
| Methods | ID | OOD | ID | OOD |
| **Standard Deep Learning** | | | | |
| NeuralODE (Chen et al., 2018) | 0.012 (0.002) | 1.098 (0.108) | 0.193 (0.024) | 1.056 (0.141) |
| DyAd (Wang et al., 2021b) | 0.016 (0.003) | 1.213 (0.216) | 0.244 (0.025) | 1.088 (0.373) |
| CoDA (Kirchmeyer et al., 2022) | NaN* | NaN* | NaN* | NaN* |
| **Physics-informed Machine Learning** | | | | |
| APHYNITY (Yin et al., 2021) | 0.047 (0.019) | 0.301 (0.139) | 0.421 (0.332) | 3.937 (1.686) |
| SINDy (Brunton et al., 2016) | NaN* | NaN* | NaN* | NaN* |
| EQL (Martius & Lampert, 2016) | NaN* | NaN* | NaN* | NaN* |
| MetaPhysiCa **(Ours)** | 0.008 (0.001) | **0.008 (0.000)** | 0.049 (0.008) | **0.129 (0.030)** |

(a) Test NRMSE ↓ for different methods. NaN* indicates that the model returned errors during test.

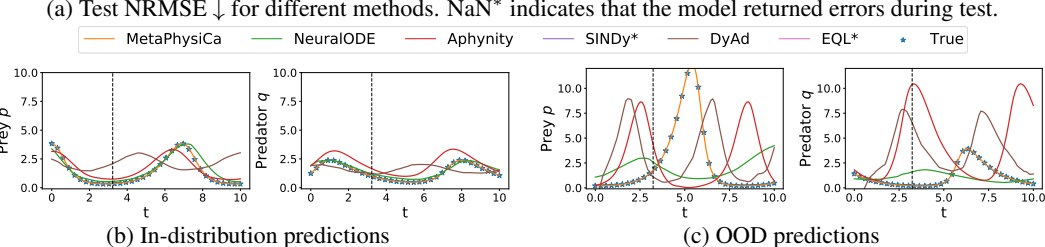

(b) In-distribution predictions          (c) OOD predictions

Figure 4: **(Predator-prey results) (a) MetaPhysiCa outputs** $30\times$ **and** $8\times$ **more robust OOD predictions** in constant $\boldsymbol{W}^{(i)*}$ and varying $\boldsymbol{W}^{(i)*}$ datasets respectively. **(b, c)** shows example ground truth curves (blue stars) in- and out-of-distribution along with corresponding predictions. While most tested methods perform well in-distribution, only MetaPhysiCa (orange) closely follows the true curve OOD.

where $\hat{\mathbf{X}}_t^{(M+1)} = \mathbf{X}_{t_0}^{(M+1)} + \int_{t_0}^t (\boldsymbol{W}^{(M+1)} \odot \hat{\Phi}) F(\hat{\mathbf{X}}_\tau^{(M+1)}; \hat{\boldsymbol{\xi}}) d\tau$ are the predictions obtained using the optimal values $\hat{\Phi}, \hat{\boldsymbol{\xi}}$, and $\lambda_{\boldsymbol{W}}$ is the hyperparameter chosen during training. Note the following two key aspects of the test-time adaptation in Equation (4): **(a)** Only the task-specific parameters $\boldsymbol{W}^{(M+1)}$ are adapted whereas the meta-model $\hat{\Phi}$ learnt during training is kept fixed, and **(b)** only the observations from time $t_0, \ldots, t_r$ of the given test trajectory is used to adapt the parameters $\boldsymbol{W}^{(M+1)}$. The final predictions $(\hat{\mathbf{X}}_t^{(M+1)})_{t_r}^{t_{T^{(M+1)}}}$ from the model are obtained with the test-time adapted parameters $\hat{\boldsymbol{W}}^{(M+1)}$ and the fixed parameters with no adaptation $\hat{\Phi}, \hat{\boldsymbol{\xi}}$.

## 5 EMPIRICAL EVALUATION

We evaluate **MetaPhysiCa** in synthetic forecasting tasks based on 3 different dynamical systems (ODEs) from the literature (Yin et al., 2021; Wang et al., 2021a) adapted to our OOD scenario, namely, **(i)** Damped pendulum system, **(ii)** Predator-prey system and **(iii)** Epidemic model. We compare against the following approaches: **(a) NeuralODE** (Chen et al., 2018), a deep learning method for learning ODEs, **(b) DyAd** (Wang et al., 2021b) (modified for ODEs), a meta-learning framework that adapts the forecaster to different training tasks with a weakly-supervised encoder, **(c) CoDA** (Kirchmeyer et al., 2022), that learns to modify its parameters to each environment with a low-rank adaptation, **(d) APHYNITY** (Yin et al., 2021), a PIML method that augments a known incomplete physics model with a neural network, **(e) SINDy** (Brunton et al., 2016), a *transductive* PIML method that uses sparse regression to learn linear coefficients over a given set of basis functions, **(f) EQL** (Martius & Lampert, 2016), a *transductive* PIML method that uses $\sin, \cos$ and other activation functions within a neural network and learns a sparse model. Additional details about the models is presented in Appendix B.

**Dataset generation.** As per Definition 1, for each dynamical system, we simulate the respective ODE to generate $M = 1000$ training tasks each observed over regularly-spaced discrete time steps $\{t_0, \ldots, t_T\}^1$ where $\forall l, t_l = 0.1l$. For each training task $\mathcal{T}^{(i)}, i = 1, \ldots, M$, we sample an initial condition $\mathbf{X}_{t_0}^{(i)} \sim P(\mathbf{X}_{t_0} | E = e)$ where $E = e$ is the training environment. We consider two scenarios for the dynamical system parameters: **(a)** *Constant* $\boldsymbol{W}^{(i)*}$, where $\boldsymbol{W}^{(i)*}$ is constant for all tasks $i$, and **(b)** *Varying* $\boldsymbol{W}^{(i)*}$, where we sample a different $\boldsymbol{W}^{(i)*} \sim P(\boldsymbol{W}^*)$ for each task $i$. Note however that none of the models have **oracle knowledge** of which of the two scenarios the data is

---

[1]In our experiments, we let $T^{(i)} = T$ constant for all tasks for simplicity of implementation but the proposed method is not restricted to this case.

observed from. At OOD test, we generate $M' = 200$ test tasks by simulating the respective dynamical system over timesteps $\{t_0, \ldots, t_r\}$, where again $\forall l, t_l = 0.1l$. For each test task $j = 1, \ldots, M'$, we sample OOD initial conditions $\mathbf{X}_{t_0}^{(j)} \sim P(\mathbf{X}_{t_0}|E = e')$ where $E = e'$ is the test environment and can induce a completely different support for the initial conditions $\mathbf{X}_{t_0}^{(j)}$ than in training. The distribution of the dynamical system parameters $\boldsymbol{W}^*$ is kept the same across training and test.

We consider three dynamical systems in our experiments, with 3 to 6 RHS terms in their respective differential equations: a damped pendulum system (Yin et al., 2021), a predator-prey system (Wang et al., 2021a), and an epidemic (SIR) model (Wang et al., 2021a), with following OOD shifts in their initial conditions respectively: acute initial angles in training to nearly vertical initial angles in OOD test, initial prey population $10\times$ less in OOD test than in training, and initial population susceptible to a disease $10\times$ more in OOD test than in training. We generate the damped pendulum dataset with 1% zero-mean Gaussian noise and the rest with no noise to show that OOD failure of baselines is unrelated to noise: existing methods fail OOD even with clean observations. For methods that require ground truth derivatives during training, we estimate them from noisy trajectories using Total Variation Regularization (TVR) (Rudin et al., 1992; Chartrand, 2011) as done by Brunton et al. (2016). Detailed description of the datasets is presented in Appendix A and experiments with increasing amounts of noise is presented in Appendix C.4.

**Results.** We repeat our experiments 5 times with different random seeds and report in-distribution (ID) and out-of-distribution (OOD) normalized root mean squared errors (NRMSE), i.e., RMSE normalized with standard deviation of the ground truth observations. Figures 2, 4 and 5 show the errors and example predictions from all models for the three datasets respectively.

The first two columns of Tables 2d, 4a, 5a show results when $\boldsymbol{W}^{(i)*}$ is constant across tasks $i$. NeuralODE, DyAd, CoDA and APHYNITY use neural network components and are able to learn the in-distribution task well with low errors. However, the corresponding errors OOD are high as they are unable to adapt to OOD initial conditions. Example OOD predictions (Figures 2c, 4c and 7b) from these methods show that they have not learnt the true dynamics of the system. For example, for epidemic modeling (Figure 7b), most models predict trajectories very similar to training trajectories even though the number of susceptible individuals is $10\times$ higher in OOD test. SINDy and EQL cannot use the training data and are fit on the test observations alone (see Figure 1(d)). Thus, they are unable to identify an accurate analytical equation from these few observations of the test task, resulting in prediction issues due to stiff ODEs. MetaPhysiCa consistently performs the best OOD across all datasets achieving $8.5\times$ to $35\times$ lower NRMSE OOD errors respectively in the 3 datasets than the best baseline. The last two columns of Tables 2d, 4a, 5a show results for the more challenging scenario when $\boldsymbol{W}^{(i)*} \sim P(\boldsymbol{W}^*)$ is varying across tasks. The results follow the same trend and MetaPhysiCa performs best OOD across all datasets achieving $8\times$ to $28\times$ lower NRMSE OOD errors respectively in the 3 datasets than the best baseline. In Appendix C.1, we show that MetaPhysiCa learns the ground truth ODE (possibly reparameterized) for all 3 dynamical systems.

## 6 CONCLUSIONS

In this work we considered the out-of-distribution (OOD) task of forecasting a dynamical system (ODE) under new initial conditions. We showed that existing PIML methods do not perform well in these tasks and proposed MetaPhysiCa that uses a meta-learning framework to learn the causal structure for the shared dynamics across all environments, while adapting the task-specific parameters. Results on three OOD (initial condition) forecasting tasks show that MetaPhysiCa is more robust with $8\times$ to $35\times$ reduction in OOD error compared to the best competing baseline.

**Limitations & future work.** We believe that forecasting models should be robust to OOD shifts, and that our work takes a step in the right direction with several potential avenues for future research: **(i)** Partial differential equations (PDEs): Extending MetaPhysiCa to forecasting PDEs under OOD scenarios is an interesting extension that requires an expanded set of basis functions that includes differential operators (like the Laplace operator), and considering out-of-distribution boundary conditions. **(ii)** More expressive structural causal models (SCMs): Our experiment on a complex ODE task (in Appendix C.5) suggests that MetaPhysiCa with a more expressive SCM that allows for composition of basis functions is able to forecast out-of-distribution better than competing baselines, but suffers from learning stiff ODEs due to the complexity of a 2-layer learnable basis function procedure. Better optimization techniques may help alleviate this problem.

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

# Supplementary Material of "MetaPhysiCa: Causality-aware Robustness to OOD Initial Conditions in Physics-informed Machine Learning"

## A  DESCRIPTION OF TASKS

For each dynamical system, we simulate the respective ODE to generate $M = 1000$ training tasks each observed over regularly-spaced discrete time steps $\{t_0, \dots, t_T\}$ where $\forall l, t_l = 0.1l$. Our data generation process is succinctly depicted in Table 1. For each dataset, the second column shows the state variables $\mathbf{X}_t$ and the unknown parameters $\boldsymbol{W}^*$. For each training task $\mathcal{T}^{(i)}, i = 1, \dots, M$, we sample an initial condition $\mathbf{X}_{t_0}^{(i)} \sim P(\mathbf{X}_{t_0}|E = e)$ where $E = e$ is the training environment (shown under ID columns of the table). We consider two scenarios for the dynamical system parameters:

- *Constant $\boldsymbol{W}^{(i)*}$* (third and fourth columns in Table 1): $\boldsymbol{W}^{(i)*}$ is constant for all tasks $i$ . For all tasks $i$, $\boldsymbol{W}^{(i)*} = \boldsymbol{W}_{\text{param}}$ as indicated in the table.

- *Varying $\boldsymbol{W}^{(i)*}$* (final two columns in Table 1): We sample a different $\boldsymbol{W}^{(i)*} \sim \mathcal{U}(\boldsymbol{W}_{\text{param}}, 2\boldsymbol{W}_{\text{param}})$ for each task $i$ with $\boldsymbol{W}_{\text{param}}$ shown in the table.

At OOD test, we generate $M' = 200$ test tasks by simulating the respective dynamical system over timesteps $\{t_0, \dots, t_r\}$, where again $\forall l, t_l = 0.1l$. For each test task $j = 1, \dots, M'$, we sample initial conditions $\mathbf{X}_{t_0}^{(j)} \sim P(\mathbf{X}_{t_0}|E = e')$ where $E = e'$ is the test environment and can induce a completely different support for the initial conditions $\mathbf{X}_{t_0}^{(j)}$ than in training. The distribution of the dynamical system parameters $\boldsymbol{W}^*$ is kept the same across training and test.

**Damped pendulum system (Yin et al., 2021).**   The state $\mathbf{X}_t = [\theta_t, \omega_t] \in \mathbb{R}^2$ describes the angle made by the pendulum with the vertical and the corresponding angular velocity at time $t$. The true (unknown) function $\psi$ describing this dynamical system is given by $\frac{d\theta_t}{dt} = \omega_t, \frac{d\omega_t}{dt} = -\alpha^{*2}\sin(\theta_t) - \rho^*\omega_t$ where $\boldsymbol{W}^* = (\alpha^*, \rho^*)$ are the dynamical system parameters. We simulate the ODE over time steps $\{t_0, \dots, t_T\}$ with $\forall l, t_l = 0.1l, T = 100$ in training and over time steps $\{t_0, \dots, t_r\}$ in test with $r = \frac{1}{3}T$. In training, the pendulum is dropped from initial angles $\theta_{t_0}^{(i)} \sim \mathcal{U}(0, \pi/2)$ with no angular velocity, whereas in OOD test, the pendulum is dropped from initial angles $\theta_{t_0}^{(j)} \sim \mathcal{U}(\pi - 0.1, \pi)$ and angular velocity $\omega_{t_0}^{(j)} \in \mathcal{U}(-1, 0)$.

**Predator-prey system (Wang et al., 2021a).**   We wish to model the dynamics between two species acting as prey and predator respectively. We adapt the experiment by Wang et al. (2021a) to our out-of-distribution forecasting scenario according to Definition 1. Let $p$ and $q$ denote the prey and predator populations respectively. The ordinary differential equations describing the dynamical system is given by $\frac{dp}{dt} = \alpha^*p - \beta^*pq, \frac{dq}{dt} = \delta^*pq - \gamma^*q$, where $\boldsymbol{W}^* = (\alpha^*, \beta^*, \gamma^*, \delta^*)$ are the (unknown) dynamical system parameters. We simulate the ODE over time steps $\{t_0, \dots, t_T\}$ with $\forall l, t_l = 0.1l, T = 100$ in training and over time steps $\{t_0, \dots, t_r\}$ in test with $r = \frac{1}{3}T$. We generate $M = 1000$ training tasks with different initial prey and predator populations with prey $p_{t_0}^{(i)} \sim \mathcal{U}(1000, 2000)$ and predator $q_{t_0}^{(i)} \sim \mathcal{U}(10, 20)$ for each $i = 1, \dots, M$. At OOD test, we generate $M' = 200$ out-of-distribution (OOD) test tasks with different initial prey populations $p_{t_0}^{(j)} \sim \mathcal{U}(100, 200)$ but the same distribution for predator population $q_{t_0}^{(j)} \sim \mathcal{U}(10, 20)$.

**Epidemic modeling (Wang et al., 2021a).**   We adapt the experiment by Wang et al. (2021a) to our out-of-distribution forecasting scenario according to Definition 1. The state of the dynamical system is described by three variables: number of susceptible ($S$), infected ($I$) and recovered ($R$) individuals. The dynamics is described using the following ODEs: $\frac{dS}{dt} = -\beta\frac{SI}{N}, \frac{dI}{dt} = \beta\frac{SI}{N} - \gamma I, \frac{dR}{dt} = \gamma I$, where $\boldsymbol{W} = (\beta, \gamma)$ are the (unknown) dynamical system parameters and $N = S + I + R$ is the total population. We simulate the ODE over time steps $\{t_0, \dots, t_T\}$ with $\forall l, t_l = 0.1l, T = 100$ in training and over time steps $r = \frac{1}{10}T$. We generate $M = 1000$ training tasks with different initial populations for susceptible ($S$) and infected ($I$) individuals, while the number of initial recovered

| Methods | Test Normalized RMSE (NRMSE) $\downarrow$ | | | |
| | Constant $\boldsymbol{W}^{(i)*}$ | | Varying $\boldsymbol{W}^{(i)*}$ | |
| | ID | OOD | ID | OOD |
|---|---|---|---|---|
| **Standard Deep Learning** | | | | |
| NeuralODE (Chen et al., 2018) | 0.003 (0.000) | 1.229 ( 0.059) | 0.005 (0.000) | 1.139 ( 0.031) |
| DyAd (Wang et al., 2021b) | 0.003 (0.001) | 121.236 (462.022) | 0.006 (0.001) | 77.377 (204.435) |
| CoDA (Kirchmeyer et al., 2022) | 0.004 (0.001) | 2.044 ( 0.754) | 0.004 (0.001) | 3.341 ( 0.389) |
| **Physics-informed Machine Learning** | | | | |
| APHYNITY (Yin et al., 2021) | 0.075 (0.071) | 2.345 ( 6.035) | 0.151 (0.150) | 0.544 ( 0.249) |
| SINDy (Brunton et al., 2016) | 2.038 (0.008) | 2.447 ( 0.065) | 1.999 (0.046) | 2.746 ( 0.476) |
| EQL (Martius & Lampert, 2016) | NaN* | NaN* | NaN* | NaN* |
| MetaPhysiCa **(Ours)** | 0.006 (0.002) | **0.035 ( 0.028)** | 0.009 (0.004) | **0.019 ( 0.002)** |

(a) Test NRMSE $\downarrow$ for different methods. NaN* indicates that the model returned errors during test.

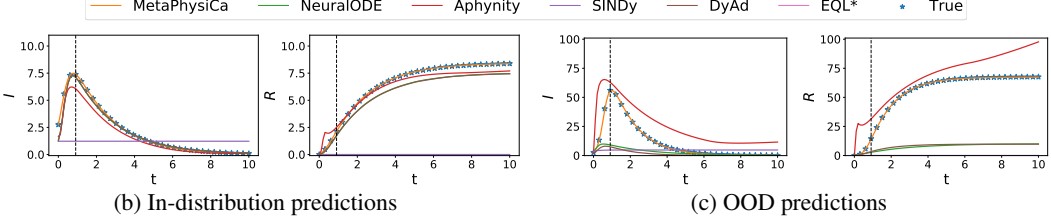

(b) In-distribution predictions        (c) OOD predictions

Figure 5: **(Epidemic model results) (a) MetaPhysiCa outputs** $35\times$ **and** $28\times$ **more robust OOD predictions** in constant $\boldsymbol{W}^{(i)*}$ and varying $\boldsymbol{W}^{(i)*}$ datasets respectively. **(b, c)** shows example ground truth curves (blue stars) in- and out-of-distribution along with corresponding predictions. Only MetaPhysiCa (orange) closely follows the true curve OOD.

| Datasets | State variables | Constant $\boldsymbol{W}^{(i)*} = \boldsymbol{W}_{\text{param}}$ | | Varying $\boldsymbol{W}^{(i)*} \sim \mathcal{U}(\boldsymbol{W}_{\text{param}}, 2\boldsymbol{W}_{\text{param}})$ | |
| | | ID | OOD | ID | OOD |
|---|---|---|---|---|---|
| Damped pendulum | $\mathbf{X}_t = (\theta_t, \omega_t)$ $\boldsymbol{W}^* = (\alpha, \rho)$ | $\theta_0 \sim \mathcal{U}(0, \pi/2)$ $\omega_0 = 0$ | $\theta_0 \sim \mathcal{U}(\pi - 0.1, \pi)$ $\omega_0 \sim \mathcal{U}(-1, 0)$ $\alpha_{\text{param}} = 1, \rho_{\text{param}} = 0.2$ | $\theta_0 \sim \mathcal{U}(0, \pi/2)$ $\omega_0 = 0$ | $\theta_0 \sim \mathcal{U}(\pi - 0.1, \pi)$ $\omega_0 \sim \mathcal{U}(-1, 0)$ |
| Predator prey system | $\mathbf{X}_t = (p_t, q_t)$ $\boldsymbol{W}^* = (\alpha, \beta, \gamma, \delta)$ | $p_0 \sim \mathcal{U}(1000, 2000)$ $q_0 \sim \mathcal{U}(10, 20)$ | $p_0 \sim \mathcal{U}(100, 200)$ $q_0 \sim \mathcal{U}(10, 20)$ $\alpha_{\text{param}} = 1, \beta_{\text{param}} = 0.06, \gamma_{\text{param}} = 0.5, \delta_{\text{param}} = 0.0005$ | $p_0 \sim \mathcal{U}(1000, 2000)$ $q_0 \sim \mathcal{U}(10, 20)$ | $p_0 \sim \mathcal{U}(100, 200)$ $q_0 \sim \mathcal{U}(10, 20)$ |
| Epidemic modeling | $\mathbf{X}_t = (S_t, I_t, R_t)$ $\boldsymbol{W}^* = (\beta, \gamma)$ | $S_0 \sim \mathcal{U}(9, 10)$ $I_0 \sim \mathcal{U}(1, 5)$ $R_0 = 0$ | $S_0 \sim \mathcal{U}(90, 100)$ $I_0 \sim \mathcal{U}(1, 5)$ $R_0 = 0$ $\beta_{\text{param}} = 4, \gamma_{\text{param}} = 0.4$ | $S_0 \sim \mathcal{U}(9, 10)$ $I_0 \sim \mathcal{U}(1, 5)$ $R_0 = 0$ | $S_0 \sim \mathcal{U}(90, 100)$ $I_0 \sim \mathcal{U}(1, 5)$ $R_0 = 0$ |

Table 1: Description of the dataset generation process. For each dataset, $\mathbf{X}_t$ denotes the state variable of the dynamical system and $\boldsymbol{W}^*$ denotes its parameters. Third and fourth columns correspond to the case when the (hidden) ground truth parameters $\boldsymbol{W}^{(i)*}$ are kept fixed for all the tasks to $\boldsymbol{W}^{(i)*} = \boldsymbol{W}_{\text{param}}$. For example, in the damped pendulum dataset, we fix $\beta^{(i)*} = \beta_{\text{param}} = 1$ and $\rho^{(i)*} = \rho_{\text{param}} = 0.2$ for all tasks $i$. Column ID represents in-distribution initial states while the column OOD represents the out-of-distribution initial states. Similarly, the final two columns correspond to the case when the ground truth parameters $\boldsymbol{W}^{(i)*}$ vary across tasks and are sampled from a uniform distribution $\boldsymbol{W}^{(i)*} \sim \mathcal{U}(\boldsymbol{W}_{\text{param}}, 2\boldsymbol{W}_{\text{param}})$. For example, in the damped pendulum dataset, we sample $\alpha^{(i)*} \sim \mathcal{U}(\alpha_{\text{param}}, 2\alpha_{\text{param}}) = (1, 2)$ and $\rho^{(i)*} \sim \mathcal{U}(\rho_{\text{param}}, 2\rho_{\text{param}}) = (0.2, 0.4)$ for each task $i$.

$(R)$ individuals are always zero. In training, we sample $S_{t_0}^{(i)} \sim \mathcal{U}(9, 10)$ and $I_{t_0}^{(i)} \sim \mathcal{U}(1, 5)$ for each $i = 1, \ldots, M$. At OOD test, we generate $M' = 200$ out-of-distribution test tasks with a different initial susceptible population, $S_{t_0}^{(j)} \sim \mathcal{U}(90, 100)$, while keeping the same distribution for infected population.

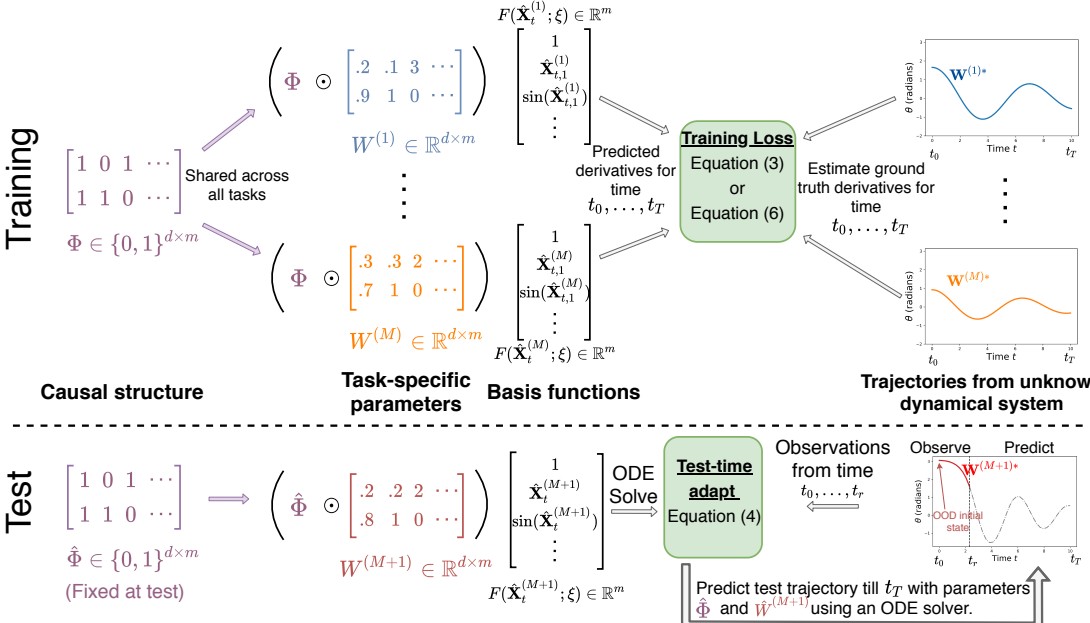

Figure 6: Schematic diagram of MetaPhysiCa and corresponding training/test methodologies. We observe $M$ trajectories in training from the same dynamical system with different initial conditions and ODE parameters. In training, $\Phi$, denoting the causal structure, is shared among all tasks $i = 1, \ldots, M$, while $\boldsymbol{W}^{(i)}$ are the task-specific parameters. Predicted derivatives for task $i$ over time $t = t_0, \ldots, t_T$ are obtained from Equation (2) using the parameters $\Phi, \boldsymbol{W}^{(i)}$ and the basis functions $F(\mathbf{X}_t^{(i)}; \boldsymbol{\xi})$. During test, we adapt $\boldsymbol{W}^{(M+1)}$ over the observations of the test trajectory from time $t_0, \ldots, t_r$, keeping the learnt causal structure $\hat{\Phi}$ fixed.

# B  IMPLEMENTATION DETAILS

In what follows, we describe implementation details of MetaPhysiCa and the baselines.

## B.1  METAPHYSICA

Figure 6 shows a schematic diagram of MetaPhysiCa and the corresponding training/test procedures. Recall from Equation (2) that the proposed model is defined as

$$\frac{d\hat{\mathbf{X}}_t^{(i)}}{dt} = (\boldsymbol{W}^{(i)} \odot \Phi) F(\hat{\mathbf{X}}_t^{(i)}; \boldsymbol{\xi}) , \tag{5}$$

where $\odot$ is the Hadamard product and

- $F(\hat{\mathbf{X}}_t^{(i)}; \boldsymbol{\xi}) := \left[ f_1(\hat{\mathbf{X}}_t^{(i)}; \boldsymbol{\xi}_1) \quad \cdots \quad f_m(\hat{\mathbf{X}}_t^{(i)}; \boldsymbol{\xi}_m) \right]^T$ is the vector of outputs from the basis functions with parameters $\boldsymbol{\xi}$,
- $\Phi \in \{0, 1\}^{d \times m}$ are the learnable parameters governing the global causal structure across all tasks such that $\Phi_{j,k} = 1$ iff edge $z_{k,t} \to {}^{d\boldsymbol{x}_{t,j}}/dt$ exists,
- $\boldsymbol{W}^{(i)} \in \mathbb{R}^{d \times m}$ are task-specific parameters that act as coefficients in linear combination of the selected basis functions.

In our experiments, we use polynomial and trigonometric basis functions, such that

$F(\hat{\mathbf{X}}_t^{(i)}; \boldsymbol{\xi}) :=$

$$\left[ 1 \quad \underbrace{\hat{\mathbf{X}}_{t,1}^{(i)} \ldots \hat{\mathbf{X}}_{t,d}^{(i)}}_{\text{polynomial order 1}} \quad \underbrace{\hat{\mathbf{X}}_{t,1}^{(i)2} \ldots \hat{\mathbf{X}}_{t,l-1}^{(i)} \hat{\mathbf{X}}_{t,l}^{(i)} \ldots \hat{\mathbf{X}}_{t,d}^{(i)2}}_{\text{polynomial order 2}} \quad \underbrace{\sin(\xi_{1,1}\hat{\mathbf{X}}_{t,1}^{(i)} + \xi_{1,2}) \ldots \sin(\xi_{d,1}\hat{\mathbf{X}}_{t,d}^{(i)} + \xi_{d,2})}_{\text{trigonometric}} \right]^T .$$

Equation (3) describes a bi-level objective that optimizes the structure parameters $\Phi$ and the global parameters $\xi$ in the outer-level, and the task-specific parameters $\boldsymbol{W}^{(i)}$ in the inner-level as follows

$$\hat{\Phi}, \hat{\boldsymbol{\xi}} = \underset{\Phi, \boldsymbol{\xi}}{\arg\min} \frac{1}{M} \sum_{i=1}^{M} R^{(i)}(\hat{\boldsymbol{W}}^{(i)}, \Phi, \boldsymbol{\xi}) + \lambda_\Phi ||\Phi||_1 + \lambda_{\text{REx}} \text{Variance}(\{R^{(i)}(\hat{\boldsymbol{W}}^{(i)}, \Phi, \boldsymbol{\xi})\}_{i=1}^{M})$$

$$\text{s.t. } \hat{\boldsymbol{W}}^{(i)} = \underset{\boldsymbol{W}^{(i)}}{\arg\min} R^{(i)}(\boldsymbol{W}^{(i)}, \Phi, \boldsymbol{\xi}) + \lambda_{\boldsymbol{W}} ||\boldsymbol{W}^{(i)}||_1, \quad \forall i = 1, \dots, M,$$

where $\lambda_\Phi$, $\lambda_{\boldsymbol{W}}$ and $\lambda_{\text{REx}}$ are hyperparameters. As discussed in the main text, the jointly optimizing $\Phi, \boldsymbol{\xi}$ and $\boldsymbol{W}^{(i)}, i = 1, \dots, M$, instead of alternating SGD resulted in comparable performance with considerable computational benefits. We use the following joint optimization objective to approximate Equation (3),

$$\hat{\Phi}, \hat{\boldsymbol{\xi}}, \hat{\boldsymbol{W}}^{(1)}, \dots, \hat{\boldsymbol{W}}^{(M)} = \underset{\Phi, \boldsymbol{\xi}, \boldsymbol{W}^{(1)}, \dots, \boldsymbol{W}^{(M)}}{\arg\min} \frac{1}{M} \sum_{i=1}^{M} R^{(i)}(\boldsymbol{W}^{(i)}, \Phi, \boldsymbol{\xi}) + \lambda_\Phi ||\Phi||_1 + \lambda_{\boldsymbol{W}} \sum_{i=1}^{M} ||\boldsymbol{W}^{(i)}||_1$$

$$\tag{6}$$

$$+ \lambda_{\text{REx}} \text{Variance}(\{R^{(i)}(\boldsymbol{W}^{(i)}, \Phi, \boldsymbol{\xi})\}_{i=1}^{M})$$

We perform a grid search over the following hyperparameters: regularization strengths $\lambda_\Phi \in \{10^{-4}, 10^{-3}, 5 \times 10^{-3}, 10^{-2}\}$, $\lambda_{\boldsymbol{W}} \in \{0, 10^{-4}, 10^{-3}, 10^{-2}\}$, $\lambda_{\text{REx}} \in \{0, 10^{-3}, 10^{-2}\}$, and learning rates $\eta \in \{10^{-2}, 10^{-3}, 10^{-4}\}$. We choose the hyperparameters that result in sparsest model (i.e., with the least $||\hat{\Phi}||_0$) while achieving validation loss within 5% of the best validation loss in held-out *in-distribution* validation data.

## B.2 NEURALODE (CHEN ET AL., 2018)

The prediction dynamics corresponding to the latent NeuralODE model is given by $\frac{d\hat{\mathbf{X}}_t}{dt} = F_{\text{nn}}(\hat{\mathbf{X}}_t, \boldsymbol{z}_{\leq r}; \boldsymbol{W}_1)$ where $\boldsymbol{z}_{\leq r} = F_{\text{enc}}(\mathbf{X}_{t_0}, \dots, \mathbf{X}_{t_r}; \boldsymbol{W}_2)$ encodes the initial observations using a recurrent neural network $F_{\text{enc}}$ (e.g., GRU), and $F_{\text{nn}}$ is a feedforward neural network. The model is trained with an ODE solver (dopri5) and the gradients computed using the adjoint method (Chen et al., 2018). We perform a grid search over the following hyperparameters: number of layers for $F_{\text{nn}}$, $L \in \{1, 2, 3\}$, size of each hidden layer of $F_{\text{nn}}$, $d_h \in \{32, 64, 128\}$, size of the encoder representation $\boldsymbol{z}_{\leq r}$, $d_z \in \{32, 64, 128\}$, batch sizes $B \in \{32, 64\}$, and learning rates $\eta \in \{10^{-2}, 10^{-3}, 10^{-4}\}$.

## B.3 DYAD (MODIFIED FOR ODES) (WANG ET AL., 2021B)

DyAd, originally proposed for forecasting PDEs, uses a meta-learning framework to adapt to different training tasks by learning a per-task weak label. We modify their approach for our ODE-based experiments. Since we do not assume the presence of weak labels for supervision for adaptation, we use mean of each variable in the training task as the task's weak label. We use NeuralODE as the base sequence model for the forecaster network. The forecaster network takes the initial observations as input and forecasts the future observations while being adapted with the encoder network. The encoder network is a recurrent network (GRU in our experiments) that takes as input the initial observations and predicts the weak label. The last layer representation from the encoder network is used to adapt NeuralODE via AdaIN (Huang & Belongie, 2017). We perform a grid search over the following hyperparameters: size of hidden layers for the forecaster and encoder networks $d_h \in \{32, 64, 128\}$, number of layers for the forecaster network, $L \in \{1, 2, 3\}$, batch sizes $B \in \{32, 64\}$, and learning rates $\eta \in \{10^{-2}, 10^{-3}, 10^{-4}\}$.

## B.4 APHYNITY (YIN ET AL., 2021)

APHYNITY assumes that we are given a (possibly incomplete) physics model $\phi(\cdot, \Theta_{\text{phy}})$ with parameters $\Theta_{\text{phy}}$. When the training data may consist of tasks with different $\boldsymbol{W}^{(i)}*$, APHYNITY predicts the physics parameters with respect to the task $i$ inductively using a recurrent neural network $G_{\text{nn}}$ from the initial observations of the system as $\hat{\Theta}_{\text{phy}}^{(i)} = G_{\text{nn}}(\mathbf{X}_{t_0}, \dots, \mathbf{X}_{t_r}; \boldsymbol{W}_2)$. Then,

APHYNITY augments the given physics model $\phi$ with a feedforward neural network component $F_{\text{nn}}$ and defines the final dynamics as $\frac{d\hat{\mathbf{X}}_t^{(i)}}{dt} = \phi(\hat{\mathbf{X}}_t^{(i)}; \hat{\Theta}_{\text{phy}}^{(i)}) + F_{\text{nn}}(\hat{\mathbf{X}}_t^{(i)}; \boldsymbol{W}_1)$. APHYNITY solves a constrained optimization problem to minimize the norm of the neural network component while still predicting the training trajectories accurately. The model is trained with an ODE solver (dopri5) and the gradients computed using the adjoint method (Chen et al., 2018). In our experiments, we provide APHYNITY with simpler physics models:

- For damped pendulum system, we use a physics model that assumes no friction: $\frac{d\theta_t}{dt} = \omega_t, \frac{d\omega_t}{dt} = -\alpha_{\text{phy}}^2 \sin(\theta_t)$ where $\Theta_{\text{phy}} = \alpha_{\text{phy}}$ is the physics model parameter.
- For predator-prey system, we use a physics model that assumes no interaction between the two species: $\frac{dp}{dt} = \alpha_{\text{phy}} p$, $\frac{dq}{dt} = -\gamma_{\text{phy}} q$ where $\Theta_{\text{phy}} = (\alpha_{\text{phy}}, \gamma_{\text{phy}})$ are the physics model parameters.
- For epidemic model, we use a physics model that assumes the disease is not infectious: $\frac{dS}{dt} = 0, \frac{dI}{dt} = -\gamma I, \frac{dR}{dt} = \gamma I$, where $\Theta_{\text{phy}} = \gamma_{\text{phy}}$ is the physics model parameter.

In each dataset, APHYNITY needs to augment the physics model with a neural network component for accurate predictions.

We perform a grid search over the following hyperparameters: number of layers for $F_{\text{nn}}$, $L \in \{1, 2, 3\}$, size of each hidden layer of $F_{\text{nn}}$, $d_h \in \{32, 64, 128\}$, batch sizes $B \in \{32, 64\}$, and learning rates $\eta \in \{10^{-2}, 10^{-3}, 10^{-4}\}$.

### B.5 SINDY (BRUNTON ET AL., 2016)

SINDy uses a given dictionary of basis functions to model the dynamics as $\frac{d\hat{\mathbf{X}}_t}{dt} = \Theta(\hat{\mathbf{X}}_t)\boldsymbol{W}$ where $\Theta$ is feature map with the basis functions (such as polynomial and trigonometric functions) and $\boldsymbol{W}$ is simply a weight matrix. SINDy is trained using sequential threshold least squares (STLS) for sparse weights $\boldsymbol{W}$. We perform a grid search over the following hyperparameters: threshold parameter used in STLS optimization, $\tau_0 \in \{0.005, 0.01, 0.05, 0.1, 0.2, 0.5\}$, and the regularization strength $\alpha \in \{0.05, 0.01, 0.1, 0.5\}$.

### B.6 EQUATION LEARNER (MARTIUS & LAMPERT, 2016)

Equation learner (EQL) is a neural network architecture where each layer is defined as follows with input $\boldsymbol{x}$ and output $\boldsymbol{o}$

$$\boldsymbol{z} = \boldsymbol{W}\boldsymbol{x} + \boldsymbol{b}$$
$$\boldsymbol{o} = (f_1(z_1), f_2(z_2), \dots, g_1(z_k, z_{k+1}), g_2(z_{k+2}, z_{k+3}), \dots,),$$

where $f_i$ are unary basis functions (such as $\sin$, $\cos$, etc.) and $g_i$ are binary basis functions (such as multiplication). We use $\text{id}, \sin$ and multiplication functions in our implementation. EQL is trained using a sparsity inducing $\ell_1$-regularization with hard thresholding for the final few epochs. We perform a grid search over the following hyperparameters: number of EQL layers, $L \in \{1, 2\}$, number of nodes for each type of basis function, $h \in \{1, 3, 5\}$, regularization strength $\alpha \in \{10^{-1}, 10^{-2}, 10^{-3}, 10^{-4}, 10^{-5}\}$, batch sizes $B \in \{32, 64\}$, and learning rates $\eta \in \{10^{-2}, 10^{-3}, 10^{-4}\}$.

## C ADDITIONAL RESULTS

### C.1 QUALITATIVE ANALYSIS

Recall from Equation (2) that the proposed model is defined as

$$\frac{d\hat{\mathbf{X}}_t^{(i)}}{dt} = (\boldsymbol{W}^{(i)} \odot \Phi) F(\hat{\mathbf{X}}_t^{(i)}; \boldsymbol{\xi}), \tag{7}$$

where $F(\hat{\mathbf{X}}_t^{(i)}; \boldsymbol{\xi})$ is the vector of outputs from the basis functions, $\Phi \in \{0, 1\}^{d \times m}$ are the learnable parameters governing the global causal structure across all tasks, and $\boldsymbol{W}^{(i)} \in \mathbb{R}^{d \times m}$ are task-specific parameters that act as coefficients in linear combination of the selected basis functions.

| Datasets | State variables | Ground truth ODE | Learnt ODE (from $\Phi$) |
|---|---|---|---|
| Damped pendulum | $\mathbf{X}_t = (\theta_t, \omega_t)$ | $\frac{d\theta_t}{dt} = \omega_t$ 
 $\frac{d\omega_t}{dt} = -\alpha^{*2}\sin(\theta_t) - \rho^*\omega_t$ | $\frac{d\theta_t}{dt} = W_1\omega_t$ 
 $\frac{d\omega_t}{dt} = W_2\sin(\theta_t) + W_3\omega_t$ |
| Predator prey system | $\mathbf{X}_t = (p_t, q_t)$ | $\frac{dp_t}{dt} = \alpha^* p_t - \beta^* p_t q_t$ 
 $\frac{dq_t}{dt} = \delta^* p_t q_t - \gamma^* q_t$ | $\frac{dp_t}{dt} = W_1 p_t + W_2 p_t q_t$ 
 $\frac{dq_t}{dt} = W_3 p_t q_t + W_4 q_t$ |
| Epidemic modeling | $\mathbf{X}_t = (S_t, I_t, R_t)$ | $\frac{dS_t}{dt} = -\beta^*\frac{S_t I_t}{S_t + I_t + R_t}$ 
 $\frac{dI_t}{dt} = \beta^*\frac{S_t I_t}{S_t + I_t + R_t} - \gamma^* I_t$ 
 $\frac{dR_t}{dt} = \gamma^* I_t$ | $\frac{dS_t}{dt} = W_1 S_t I_t$ 
 $\frac{dI_t}{dt} = W_2 S_t I_t + W_3 I_t^2 + W_4 I_t R_t$ 
 $\frac{dR_t}{dt} = W_5 S_t I_t + W_6 I_t^2 + W_7 I_t R_t$ |

Table 2: **(Qualitative analysis.)** Ground truth dynamical system vs learnt ODE in the meta-model $\Phi$. Recall that $\Phi \in \{0, 1\}^{d \times m}$ dictates which of the basis functions affect the output $d\mathbf{X}_t/dt$. The weights $W_l$ in the learnt ODE column are learnable parameters that are optimized via test-time adaptation in Equation (4). **MetaPhysiCa learns the exact ground truth ODE for Damped pendulum and Predator-prey system, and a reparameterized version of the true ODE for epidemic modeling task.**

After training, the ODE learnt by the model can be easily inferred by checking all the terms in $\Phi$ that are greater than zero, i.e., $\Phi_{j,k} > 0$ implies $f_k(\boldsymbol{x}_t; \boldsymbol{\xi}_k) \to d\boldsymbol{x}_{t,j}/dt$ exists in the causal graph. In other words, RHS of learnt ODE for $d\boldsymbol{x}_{t,j}/dt$ contains the basis function $f_k(\boldsymbol{x}_t; \boldsymbol{\xi}_k)$.

Table 2 shows the ground truth ODE and the learnt ODE for the three experiments. For each learnt ODE, we also depict the learnable parameters $W_l$ that can be adapted using Equation (4) during test-time. For damped pendulum and predator-prey system, the RHS terms in the learnt ODE exactly matches ground truth ODE, and from Figures 2 and 4, it is clear that the method is able to accurately adapt the learnable parameters $W_l$ during test-time. For epidemic modeling task, MetaPhysiCa learns a reparameterized version of the ground truth ODE. For example, MetaPhysiCa learns $\frac{dR_t}{dt} = W_a' I_t S_t + W_b' I_t^2 + W_c' I_t R_t$, which can be written as $\frac{dR_t}{dt} = W_a I_t$ (the ground truth ODE) if $W_a' = W_b' = W_c'$, because $S_t + I_t + R_t = N$ is a constant denoting the total population. While the learnt reparameterized ODE is more complex because it allows different values for $W_a', W_b', W_c'$, the test-time adaptation of these learnable parameters with the initial test observations results in them taking the same values.

## C.2 Ablation results

We present an ablation study comparing different components of MetaPhysiCa in Table 3. Table shows out-of-distribution test NRMSE for MetaPhysiCa without each individual component on the three dynamical systems (varying $\boldsymbol{W}^{(i)*}$ scenario). We observe that sparsity regularization (i.e., $||\Phi||_1$) and test-time adaptation are the most important components. For two out of three tasks, the method returns prediction errors without sparsity regularization.

When testing MetaPhysiCa without test-time adaptation, we simply use the mean of the task-specific weights learnt for training tasks as the task-specific weight for the given test trajectory, i.e., $\hat{\boldsymbol{W}}^{M+1} = \frac{1}{M}\sum_i \boldsymbol{W}^{(i)}$. This results in high OOD errors showing the importance of test-time adaptation. The other two components of the MetaPhysiCa, the task-specific $\ell_1$-regularization (i.e., $||\boldsymbol{W}^{(i)}||_1$) and the V-REx penalty (Krueger et al., 2021) help in some experiments and perform comparably in others.

## C.3 Out-of-distribution ODE parameters

The forecasting task in Definition 1 considers out-of-distribution initial conditions $\mathbf{X}_{t_0}$ and in-distribution ODE parameters $\boldsymbol{W}^{(i)*}$. Here, we consider OOD values for true dynamical system parameters $\boldsymbol{W}^{(i)*}$ as well, which significantly increases the difficulty of the forecasting task.

Consider the damped pendulum system: $\frac{d\theta_t}{dt} = \omega_t, \frac{d\omega_t}{dt} = -\alpha^{*2}\sin(\theta_t) - \rho^*\omega_t$ where $\boldsymbol{W}^* = (\alpha^*, \rho^*)$ are the dynamical system parameters. *Training:* Pendulum is dropped from initial angles $\theta_{t_0}^{(i)} \sim \mathcal{U}(0, \pi/2)$ with no angular velocity. We sample the dynamical system parameters $\alpha^{(i)*} \sim \mathcal{U}(1, 2)$

| Method | OOD Test Normalized RMSE ↓ | | |
| --- | --- | --- | --- |
| | Damped Pendulum | Predator-Prey | Epidemic Modeling |
| MetaPhysiCa | **0.070 (0.011)** | **0.129 (0.030)** | **0.019 (0.002)** |
|   without $\|\|\Phi\|\|_1$ | NaN* | 1.806 (0.736) | NaN* |
|   without $\|\|\boldsymbol{W}^{(i)}\|\|_1$ | 0.434 (0.531) | **0.132 (0.020)** | **0.020 (0.021)** |
|   without test-time adaptation | 1.223 (0.741) | 1.404 (3.794) | 0.358 (0.554) |
|   without V-REx penalty | **0.070 (0.014)** | **0.129 (0.030)** | 0.042 (0.065) |

Table 3: **(Ablation.)** Out-of-distribution test NRMSE for MetaPhysiCa without each individual component on the three dynamical systems (varying $\boldsymbol{W}^{(i)*}$ scenario). **Sparsity regularization (i.e., $\|\|\Phi\|\|_1$) and test-time adaptation are the most important components, whereas the task-specific $\ell_1$-regularization (i.e., $\|\|\boldsymbol{W}^{(i)}\|\|_1$) and the V-REx penalty (Krueger et al., 2021) help in some tasks, but not in others.**

| Methods | Test NRMSE ↓ | | |
| --- | --- | --- | --- |
| | ID | OOD $\mathbf{X}_{t_0}$ | OOD $\mathbf{X}_{t_0}$ and $\boldsymbol{W}^{(i)*}$ |
| **Standard Deep Learning** | | | |
|  NeuralODE (Chen et al., 2018) | 0.083 (0.033) | 0.591 (0.119) | 1.208 (0.401) |
|  DyAd (Wang et al., 2021b) | 0.078 (0.051) | 0.834 (0.263) | 1.390 (0.441) |
|  CoDA (Kirchmeyer et al., 2022) | 0.052 (0.032) | 0.764 (0.201) | 1.031 (0.213) |
| **Physics-informed Machine Learning** | | | |
|  APHYNITY (Yin et al., 2021) | 0.097 (0.020) | 0.970 (0.384) | 1.343 (0.404) |
|  SINDy (Brunton et al., 2016) | NaN* | NaN* | NaN* |
|  EQL (Martius & Lampert, 2016) | NaN* | NaN* | NaN* |
|  MetaPhysiCa **(ours)** | 0.049 (0.002) | **0.070 (0.011)** | **0.181 (0.012)** |

Table 4: **(Damped pendulum.)** Normalized RMSE ↓ of test predictions from different methods under two cases: **(a)** when initial conditions $\mathbf{X}_{t_0}$ are OOD, and **(b)** when both initial conditions $\mathbf{X}_{t_0}$ and ODE parameters $\boldsymbol{W}^{(i)*}$ are OOD. NaN* indicates that the model returned errors during test-time predictions. **MetaPhysiCa is able to adapt its parameters to the OOD parameters $\boldsymbol{W}^{(i)*}$ and outputs $\approx 5\times$ more robust OOD predictions compared to the baselines.**.

and $\rho^{(i)*} \sim \mathcal{U}(0.2, 0.4)$ for each task $i$. *Out-of-distribution Test:* The pendulum is dropped from initial angles $\theta_{t_0}^{(j)} \sim \mathcal{U}(\pi - 0.1, \pi)$ and angular velocity $\omega_{t_0}^{(j)} \in \mathcal{U}(-1, 0)$. We sample the dynamical system parameters $\alpha^{(i)*} \sim \mathcal{U}(1, 2)$ and $\rho^{(i)*} \sim \mathcal{U}(0.1, 0.2)$. Note that the damping coefficient $\rho^{(i)*}$ is sampled out-of-support from its training distribution. Rest of the experimental methodology is kept same as before.

We report the normalized RMSE of the all the methods in Table 4 for three test scenarios: in-distribution (ID), out-of-distribution initial conditions (OOD $\mathbf{X}_{t_0}$), and out-of-distribution initial conditions and ODE parameters (OOD $\mathbf{X}_{t_0}$ and $\boldsymbol{W}^{(i)*}$). MetaPhysiCa is able to adapt relatively well to the out-of-distribution ODE parameters and performs $\approx 4\times$ better than the best baseline. Unfortunately, the test-time adaptation is not perfect (NRMSE is $5\times$ higher for OOD $\mathbf{X}_{t_0}$ and $\boldsymbol{W}^{(i)*}$ compared to OOD initial conditions alone), possibly because the trajectories with higher $\alpha^{(i)*}$ and higher $\rho^{(i)*}$ are harder to forecast.

## C.4 ROBUSTNESS TO NOISE

We repeat the Damped pendulum and Predator-prey experiments with increasing amounts of noises. Specifically, we add $1\%, 5\%$ and $10\%$ Gaussian noise to all the trajectories, both in training and in test. As discussed before, we use Total Variation Regularization (TVR) (Rudin et al., 1992; Chartrand, 2011) for estimating derivatives from noisy data as done by Brunton et al. (2016). We report the normalized RMSE for different models trained on the noisy versions of data in Figure 7. SINDy and EQL are not shown as they returned errors during test-time predictions similar to the case with no noise because the learnt ODE was too stiff (numerically unstable) to solve. In both tasks, the proposed

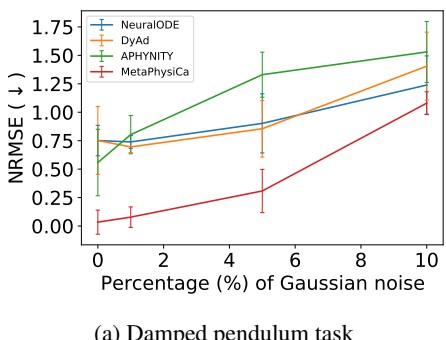
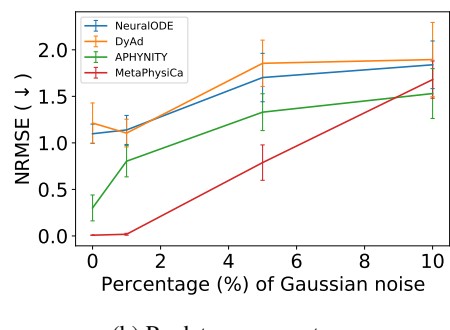

(a) Damped pendulum task                (b) Predator-prey system

Figure 7: **(Performance with increasing noise.)** Out-of-distribution NRMSE values for Damped Pendulum and Predator-prey experiments with different percentages of Gaussian noise added $(0\%, 1\%, 5\%, 10\%)$. **MetaPhysiCa is relatively robust to $\leq 5\%$ Gaussian noise and outperforms the baselines. With a larger amount of noise, MetaPhysiCa is unable to identify the dynamical system accurately but performs comparable to the baselines.**

method is relatively robust to small amounts of noise and outperforms the baselines. With $10\%$ noise, MetaPhysiCa is unable to identify the dynamical system accurately, but performs comparable to the baselines.

## C.5    COMPLEX ODE TASK

In this section, we extend MetaPhysiCa to consider significantly more expressive structural causal models (compared to Figure 3) that allow for composition of the basis functions. This is achieved with a 2-layer learnable basis function composition procedure. For example, given basis functions $f_1(\boldsymbol{x}_t; \xi_1) = \sin(\xi_{1,1}\boldsymbol{x}_{t,1} + \xi_{1,2})$, and $f_2(\boldsymbol{x}_t; \xi_2) = \boldsymbol{x}_{t,1}\boldsymbol{x}_{t,2}$, one can construct more expressive basis functions with compositions: $\tilde{f}_3(\boldsymbol{x}_t; \xi_3) = \sin(\xi_{3,3}\sin(\xi_{3,1}\mathbf{x}_{t,1} + \xi_{3,2}) + \xi_{3,4})$, $\tilde{f}_4(\boldsymbol{x}_t; \xi_4) = \boldsymbol{x}_{t,1}\boldsymbol{x}_{t,2}\sin(\xi_{4,1}\boldsymbol{x}_{t,1} + \xi_{4,2})$, etc., where $\xi_j$ are global parameters that remain constant for all training/test tasks. The rest of the SCM remains the same and the derivative $d\mathbf{x}_{t,j}^{(i)}/dt$ for a particular dimension $j \in \{1, \ldots, d\}$ is a sparse linear combination of the original basis functions and the more expressive second layer ones.

We evaluated MetaPhysiCa on a more complex ODE task from Chen (2020) adapted to our setting. We consider a two-dimensional ODE with state $\mathbf{X}_t = [p_t, q_t] \in \mathbb{R}^2$: $\frac{dp_t}{dt} = a^* \sin(p_t) + b^* \sin(q_t^2)$; $\frac{dp_t}{dt} = c^* \sin(p_t) \cos(q_t)$, where $\boldsymbol{W}^* = (a^*, b^*, c^*)$ are the dynamical system parameters. We simulate the ODE over time steps $\{t_0, \ldots, t_T\}$ with $\forall l, t_l = 0.1l, T = 100$ in training and over time steps $\{t_0, \ldots, t_r\}$ in test with $r = \frac{1}{3}T$. In training, we sample initial states $p_t, q_t \sim \mathcal{U}(0.5, 1)$, whereas in out-of-distribution test, we sample $p_t, q_t \sim \mathcal{U}(1, 1.5)$. For constant $\boldsymbol{W}^{(i)*}$ scenario, the dynamical system parameters are set to $a^{(i)*} = b^{(i)*} = c^{(i)*} = 1$ for all $i$, whereas for the varying $\boldsymbol{W}^{(i)*}$ scenario, the dynamical system parameters are sampled as $a^{(i)*}, b^{(i)*}, c^{(i)*} \sim \mathcal{U}(1.0, 1.5)$.

Table 5 shows the results for this task. First, we note that due to the complexity of a 2-layer learnable basis function procedure, we sometimes need to use validation data (held out from training) to cross-validate the learned model (and reject meta-models that do not do well in validation). MetaPhysiCa learnt a stiff ODE for 2 out of 5 folds of cross-validation, resulting in no predictions for in-distribution validation data, which were rejected (marked as superscript $*$). In these experiments MetaPhysiCa performs $1.5\times$ to $1.7\times$ better than the competing baselines. We believe there is room for improvement in the optimization procedure of these more complex models.

| | Test Normalized RMSE (NRMSE) ↓ | | | |
| | Constant $\boldsymbol{W}^{(i)*}$ | | Varying $\boldsymbol{W}^{(i)*}$ | |
| Methods | ID | OOD | ID | OOD |
|---|---|---|---|---|
| NeuralODE (Chen et al., 2018) | 0.012 (0.001) | 0.188 (0.025) | 0.034 (0.008) | 0.296 (0.064) |
| APHYNITY (Yin et al., 2021) | 0.010 (0.002) | 0.329 (0.050) | 0.027 (0.010) | 0.684 (0.117) |
| SINDy (Brunton et al., 2016) | NaN* | NaN* | NaN* | NaN* |
| MetaPhysiCa (Ours) | 0.119 (0.072)* | 0.110 (0.048)* | 0.188 (0.035)* | 0.203 (0.046)* |

Table 5: Test NRMSE ↓ for different methods. * indicates that the method returned errors during predictions due to learning a stiff ODE.

