# OpenReview forum: "MetaPhysiCa: Causality-aware Robustness to OOD Initial Conditions in Physics-informed Machine Learning"
_ICLR.cc/2023/Conference — Submitted to ICLR 2023_

### Official Review · Reviewer_hnK8 · 2022-10-24

**Confidence:** 4
**Correctness:** 3
**Technical Novelty And Significance:** 3
**Empirical Novelty And Significance:** 3
**Recommendation:** 6

**Clarity, Quality, Novelty And Reproducibility:**

The paper is clearly written, and easy to follow and the experimental evaluation seems technically rigorous and sound. Various components and assumptions of the method such as sparsity regularization or meta-learning are well-known regularization techniques but it is the combination of all these components that has some novelty to the best of my knowledge.


**Strength And Weaknesses:**

Strengths:
- The paper is very clearly written and the problem setup, as well as the proposed method, are presented comprehensively.
- The authors validate their method on three prominent ODE tasks and benchmark their method against a multitude of relevant baselines
- The paper touches upon an important problem setting and I appreciate the general idea of the method to obtain better OOD generalization via causality-inspired regularization and modelling assumptions.
- Nice and convincing experimental results
- Nice contextualization of the method with respect to prior works

Weaknesses:
- The method appears to make several rather strong assumptions on the underlying ODE system that make the method only apply to a smaller subset of ODE systems. (1) The underlying ODE system is assumed to be parametrized with the task-specific parameters W acting as coefficients in a linear combination of basis functions, which happens to be exactly the case for the studied toy experiments. I thus would expect the method to potentially exhibit several issues if this is not possible. The three environments are all building upon the sparsest possible causal structure in that every RHS is a single term such that the sparsity regularization will always help as long as exactly one entry per row in the structure parameter matrix survives. It would be important to understand how the method performs if there are linear combinations of more than one basis function in the RHS. Finding a good trade-off in the right regularization through $\lambda_\Phi$ and its effect on OOD generalization might quickly become very challenging. While I understand that the more general setting is much harder and I appreciate the progress over baselines in the here studied experiments I would have appreciated it if these limitations and assumptions are stated more prominently as it seems the method predominantly benefits a lot from being given optimal basis functions.
- Presentation (minor): The plots in Figures 4 and 5 should be enlarged and could be improved as the labels and legends are barely readable.


**Summary Of The Paper:**

This paper proposes a method called MetaPhysiCa, which is a physics-informed machine learning method to better model dynamical ODE systems tested under OOD Initial conditions with changing or non-changing ODE parameters. The approach is making use of various meta-learning and causal structure discovery techniques in combination with a predefined set of possible basis functions to parametrize the RHS of an ODE. Authors validate their approach on 3 different ODE setups (damped pendulum, predator-prey and epidemic modelling) outperforming multiple baselines in OOD test scenarios.

**Summary Of The Review:**

I think this paper is a nice contribution by proposing a new physics-informed ML method for ODEs that exhibit more robust generalization with respect to OOD initial conditions and makes use of several important concepts and necessary assumptions to achieve this. While I appreciate the number of experiments and convincing performance over existing baselines I have mentioned a few remaining concerns as the studied setups appear to be particularly easy for the here proposed method. I am concerned that the method might not be very applicable beyond such very simple ODEs that adhere to the structure assumed by the method. I, therefore, lean towards rejecting but I am willing to change my rating if the authors can address my concerns.

---

> ### Author Response · Authors · 2022-11-18
> **Response to Reviewer hnK8**
>
> We thank the reviewer for their positive comments and insightful feedback.
>
> **Q1:** The method appears to make several rather strong assumptions on the underlying ODE system that make the method only apply to a smaller subset of ODE systems. I thus would expect the method to potentially exhibit several issues if this is not possible.
>
> **A1:** Thank you for the feedback. In the updated draft we extended MetaPhysiCa to consider significantly more expressive structural causal models that allow for composition of the basis functions. This is achieved with a 2-layer learnable basis function composition procedure (detailed in Appendix C.5). We have also added an experiment on a more complex ODE task (adapted from [1]). We describe the experiment and results below briefly.
>
> Consider for example, 2 given basis functions $f_1(x_t; \xi_1) = \sin(\xi_{1,1} x_{t,1} + \xi_{1,2})$, and $f_2(x_t; \xi_2) = x_{t,1} x_{t,2}$. We construct more expressive basis functions via compositions: $\tilde{f}\_3(x_t;\xi_3)=\sin(\xi_{3,3} \sin(\xi_{3, 1} x_{t, 1} + \xi_{3,2}) + \xi_{3,4})$, $\tilde{f}\_4(x_t;\xi_4)= x_{t,1} x_{t, 2} \sin(\xi_{4,1} x_{t, 1} + \xi_{4,2})$, etc., where $\xi_j$ are global parameters that remain constant for all training/test tasks. The rest of the SCM (Figure 3) remains the same and the derivative $\frac{d{x}^{(i)}_{t, j}}{dt}$ for a particular dimension $j\in \{1,\ldots,d\}$ is a sparse linear combination of the original basis functions and the more expressive second layer ones.
>
> **Experiment:** We consider a two-dimensional ODE with state $x_t = [p_t, q_t]$: $\frac{dp_t}{dt} = a^* \sin(p_t) + b^* \sin(q_t^2); \frac{dp_t}{dt} = c^* \sin(p_t)\cos(q_t)$, where $W^*=(a^*, b^*, c^*)$ are the dynamical system parameters. In training, we sample initial states $p_t, q_t \sim \mathcal{U}(0.5, 1)$, whereas in out-of-distribution test, we sample $p_t, q_t \sim \mathcal{U}(1, 1.5)$.
>
> Table 5 shows the results for this task. First, we note that, due to the complexity of a 2-layer learnable basis function procedure, we sometimes need to use validation data (held out from training) to cross-validate the learned model (and reject models that do not do well in validation). In these experiments MetaPhysiCa performs $1.5\times$ to $1.7\times$ better than the competing baselines. We believe there is room for improvement in the optimization procedure of these more complex models.
>
>
>
> **Q2:** I would have appreciated it if these limitations and assumptions are stated more prominently as it seems the method predominantly benefits a lot from being given optimal basis functions.
>
> **A2:** We have expanded our Conclusions section with a discussion on the limitations and possible avenues for future work as follows:
>
> “We believe that forecasting models should be robust to OOD shifts, and that our work takes a step in the right direction with several potential avenues for future research:
>
> **(i)** Partial differential equations (PDEs): Extending MetaPhysiCa to forecasting PDEs under OOD scenarios is an interesting extension that requires an expanded set of basis functions that includes differential operators (like the Laplace operator), and considering out-of-distribution boundary conditions.
>
> **(ii)** More expressive structural causal models (SCMs): Our experiment on a complex ODE task (in Appendix C.5) suggests that MetaPhysiCa with a more expressive SCM that allows for composition of basis functions is able to forecast out-of-distribution better than competing baselines, but suffers from learning stiff ODEs due to the complexity of a 2-layer learnable basis function procedure. Better optimization techniques may help alleviate this problem. “
>
>
>
> **Q3:** The three environments are all building upon the sparsest possible causal structure in that every RHS is a single term such that the sparsity regularization will always help as long as exactly one entry per row in the structure parameter matrix survives. It would be important to understand how the method performs if there are linear combinations of more than one basis function in the RHS.
>
> **A3:** Thanks for the feedback. This is now emphasized in the text: The RHS of the dynamical systems tested contain *more than one* basis term. For instance, the predator-prey system is defined by: $\frac{dp}{dt} = \alpha^* p - \beta^* p q \:, \frac{dq}{dt} = \delta^* p q - \gamma^* q$, where the model needs to identify a total of four RHS terms (two for prey dynamics and two for predator dynamics).
>
>
> **Q4:** Presentation (minor): The plots in Figures 4 and 5 should be enlarged and could be improved as the labels and legends are barely readable.
>
> **A4:** Thank you, we have enlarged the plot labels and legends for better readability.
>
>
>
>
> **References:**
>
> [1] Gang Chen. Learning symbolic expressions via gumbel-max equation learner networks. arXiv preprint arXiv:2012.06921, 2020

---

> > ### Comment · Reviewer_hnK8 · 2022-11-24
> > **Response Rebuttal. Increasing my score.**
> >
> > I would like to thank the authors for their reply and appreciate their efforts in running additional experiments on more complex SCMs that resolved some of my previously mentioned concerns. I am therefore happy to increase my score and confidence.

---

### Official Review · Reviewer_s2pT · 2022-10-31

**Confidence:** 3
**Correctness:** 3
**Technical Novelty And Significance:** 3
**Empirical Novelty And Significance:** 3
**Recommendation:** 6

**Clarity, Quality, Novelty And Reproducibility:**

**Clarity**: The paper is well-written and easy to follow.
**Quality**: The authors compare against relevant baselines and present significant improvements in OOD tasks.
**Novelty**: The proposed algorithm is a novel contribution that builds on well-studied meta-learning and causal inference ideas.
**Reproducibility**: The authors present information sufficient to reproduce key-experiments and results from the paper.

**Strength And Weaknesses:**

### **Strength**
+ The paper is well-written, with pedagogical examples highlighting the limits and challenges of current PIML algorithms. The authors do a great job presenting their contributions in the context of existing literature.
+ By effectively combining the strengths of meta-learning algorithms and causal inference, the proposed algorithm significantly improves performance on OOD tasks in well-studied benchmark physics simulation tasks (such as damped pendulum and predator-prey).

### **Weakness**
- The assumptions of realizability in the family of structural causal models seem restrictive and might not hold in more complex ODE tasks. A discussion on the scope of these assumptions would significantly help understand the algorithm's limitations and strengths.

**Summary Of The Paper:**

The authors consider the challenging problem of learning robust solvers for ODEs in physics-inspired machine learning, mainly when initial conditions at the test time are the out-of-distribution. First, with simple experiments on well-motivated simulated physics tasks (such as damped pendulum systems and predator-prey), the paper presents the limitations of existing PIML algorithms in dealing with OOD initial conditions. Next, the authors propose meta-learning algorithms that embed a structural causal model to identify the most suitable model for the ODE task to tackle this. Finally, evaluating the normalized RMSE, experiments across multiple domains suggest that MetaPhysiCa is competitive on in-distribution tasks but significantly outperforms the baselines in OOD settings.

**Summary Of The Review:**

The paper studies challenges in forecasting dynamical systems with OOD initial conditions and the limits of current PIML-based approaches. Building on insights from causal inference and multi-task learning, the authors propose MetaPhysica. This hybrid algorithm performs a structural causal search on a family of SCMs (under realizability) to perform appropriate interventions. With extensive experiments and comparisons to relevant baselines, the proposed algorithm outperforms prior work on OOD tasks by a considerable margin.

---

> ### Author Response · Authors · 2022-11-18
> **Response to Reviewer s2pT**
>
> We thank the reviewer for their positive comments and feedback.
>
> **Q1:** The assumptions of realizability in the family of structural causal models seem restrictive and might not hold in more complex ODE tasks. A discussion on the scope of these assumptions would significantly help understand the algorithm's limitations and strengths.
>
>
> **A1:** Thank you for the feedback. In the updated draft we extended MetaPhysiCa to consider significantly more expressive structural causal models that allow for composition of the basis functions. This is achieved with a 2-layer learnable basis function composition procedure (detailed in Appendix C.5). We have also added an experiment on a more complex ODE task (adapted from [1]). We describe the experiment and results below briefly.
>
>
> Consider for example, 2 given basis functions $f_1(x_t; \xi_1) = \sin(\xi_{1,1} x_{t,1} + \xi_{1,2})$, and $f_2(x_t; \xi_2) = x_{t,1} x_{t,2}$. We construct more expressive basis functions via compositions: $\tilde{f}\_3(x_t;\xi_3)=\sin(\xi_{3,3} \sin(\xi_{3, 1} x_{t, 1} + \xi_{3,2}) + \xi_{3,4})$, $\tilde{f}\_4(x_t;\xi_4)= x_{t,1} x_{t, 2} \sin(\xi_{4,1} x_{t, 1} + \xi_{4,2})$, etc., where $\xi_j$ are global parameters that remain constant for all training/test tasks. The rest of the SCM (Figure 3) remains the same and the derivative $\frac{d{x}^{(i)}_{t, j}}{dt}$ for a particular dimension $j\in \{1,\ldots,d\}$ is a sparse linear combination of the original basis functions and the more expressive second layer ones.
>
> **Experiment:** We consider a two-dimensional ODE with state $x_t = [p_t, q_t]$: $\frac{dp_t}{dt} = a^* \sin(p_t) + b^* \sin(q_t^2); \frac{dp_t}{dt} = c^* \sin(p_t)\cos(q_t)$, where $W^*=(a^*, b^*, c^*)$ are the dynamical system parameters. In training, we sample initial states $p_t, q_t \sim \mathcal{U}(0.5, 1)$, whereas in out-of-distribution test, we sample $p_t, q_t \sim \mathcal{U}(1, 1.5)$.
>
> Table 5 shows the results for this task. First, we note that, due to the complexity of a 2-layer learnable basis function procedure, we sometimes need to use validation data (held out from training) to cross-validate the learned model (and reject models that do not do well in validation). In these experiments MetaPhysiCa performs $1.5\times$ to $1.7\times$ better than the competing baselines. We believe there is room for improvement in the optimization procedure of these more complex models.
>
>
> We have expanded our Conclusions section with a discussion on the limitations and possible avenues for future work as follows:
>
> “We believe that forecasting models should be robust to OOD shifts, and that our work takes a step in the right direction with several potential avenues for future research:
>
> **(i)** Partial differential equations (PDEs): Extending MetaPhysiCa to forecasting PDEs under OOD scenarios is an interesting extension that requires an expanded set of basis functions that includes differential operators (like the Laplace operator), and considering out-of-distribution boundary conditions.
>
> **(ii)** More expressive structural causal models (SCMs): Our experiment on a complex ODE task (in Appendix C.5) suggests that MetaPhysiCa with a more expressive SCM that allows for composition of basis functions is able to forecast out-of-distribution better than competing baselines, but suffers from learning stiff ODEs due to the complexity of a 2-layer learnable basis function procedure. Better optimization techniques may help alleviate this problem. “
>
>
> **References:**
>
> [1] Gang Chen. Learning symbolic expressions via gumbel-max equation learner networks. arXiv preprint arXiv:2012.06921, 2020

---

### Official Review · Reviewer_t2KB · 2022-11-02

**Confidence:** 4
**Correctness:** 4
**Technical Novelty And Significance:** 3
**Empirical Novelty And Significance:** 2
**Recommendation:** 5

**Clarity, Quality, Novelty And Reproducibility:**

This hybrid method combining meta-learning and structural causal discovery seems quite new to me, and very promising. The causality part should be more studied and interpreted (W1). What is the link with other recent approaches leveraging causality in PIML, eg [1] ?
The implementation details are missing (in paper or appendix).

What are the neural network architectures used? (a schematic of the model could be fine for understanding but is not mandatory) How are chosen the hyperparameters? How are chosen the basis functions for the SCM? Are they fixed manually or can they be learned? The neural network details should also be reported for the baselines for a fair comparison. Besides, will the code be released?

Typos:

Page 2: the definition of xi_star is missing.

Page 7: “a structure that (is) minimizes”

[1] Sifan Wang, Shyam Sankaran, Paris Perdikaris, Respecting causality is all you need for training physics-informed neural networks

**Strength And Weaknesses:**

Strengths:

This paper is well-written and very enjoyable to read. It tackles a very relevant issue that undermines physics-informed machine learning in OOD settings. The method is clearly presented and well positioned with respect to the state-of-the-art. The combination of the structural causal model with meta-learning is an appealing idea. This method outperforms with a large margin other competitors on OOD initial conditions, while remaining equivalent in-distribution.

Weaknesses:

W1: The experimental section contains only forecasting performances in and out of distribution. Without an in-depth analysis of the model, for example an ablation study, it is hard to understand what elements make this model more successful. Could you provide (possibly in appendix) a more thorough analysis? For instance, what is the impact of the SCM? Have you analyzed the discovered causal structure and what interpretation can be drawn?


W2: Moreover, considering the robustness to ODD initial conditions is fine and interesting, but why not addressing in the paper the robustness to ODE system parameters? This is done in CoDA and DyAd and I a wondering if this method would still have such a performance gain in this context. This would be interesting for the community to analyze what source of ODD interventions have the most impact.

W3: The authors make experiments on common benchmark ODEs (damped pendulum, predator prey, epidemic). Why not evaluate your method on more complex PDE equations? Published methods evaluate on 2D reaction-diffusion and Navier Stockes (for CoDA) and turbulent flows and sea surface temperature (for DyAd).

Questions:

Q1: The loss minimized in practice is a MSE between the derivatives along the trajectories. How is the derivative estimated for the ground-truth trajectory (finite differences)? Are these estimates noisy with noisy data and how does it affect performances?

Q2: In Eq 3, the authors write the global optimization objective. Is this the real bi-level objective optimized in practice? An algorithm script could help to clarify the training scheme.



**Summary Of The Paper:**

Current deep and physics-informed machine learning models struggle to correctly forecast dynamical systems in out-of-distribution (OOD) settings. This paper proposes a new approach consisting of a meta-leaning strategy combined with causal structural discovery. The method is evaluated on standard ODE benchmarks, showing better forecasting performances on OOD initial conditions.

**Summary Of The Review:**

In summary, this hybrid approach for OOD dynamical forecasting is appealing and shows promising results. However, in the current state of the paper, it is hard to really understand the working principle and if the method would apply to more complex physical systems.

EDIT: after discussion with other reviewers, I have still concerns on the clarity of the method. The positioning with respect to Coda and DyAd is not discussed and  the reason that this method outperform the other by an order of magnitude is still unclear for me. Besides, I do not understand in experiments why Aphynity is largely inferior to the data-driven Neural ODE for the in-distribution pendulum.
I return to my original pre-rebuttal score.

---

> ### Author Response · Authors · 2022-11-18
> **Response to Reviewer t2KB (Part 3/3)**
>
> **Q9:** How are chosen the hyperparameters?
>
> **A9:** We have now emphasized our hyperparameter selection strategy above section 4.3 in Page 7.
>
> “Hyperparameter selection: We choose the hyperparameters that result in sparsest model (i.e., with the least $||\hat{\Phi}||_0$) while achieving validation loss within 5% of the best validation loss in held-out in-distribution validation data. The use of in-distribution data for validation is key requirement since in OOD tasks one does not have access to samples from the test distribution.”
>
>
>
> **Q10:** How are chosen the basis functions for the SCM? Are they fixed manually or can they be learned?
>
> **A10:** We assume that we are given a set of basis functions as domain knowledge of the task similar to prior works [4, 7]. However, unlike prior works, we allow learnable global parameters for the basis functions ($\xi$ in Equation 2) that are constant across all training and test tasks. This allows for partial learning of basis functions, for example, the phase term of a sine function.
>
>
>
> **Q11:** Besides, will the code be released?
>
> **A11:** Yes, we will provide anonymized code for our proposed method.
>
> Typos: Thank you, we have fixed the typos.
>
>
>
> **References:**
>
> [1] Wang, Sifan, Shyam Sankaran, and Paris Perdikaris. "Respecting causality is all you need for training physics-informed neural networks." *arXiv preprint arXiv:2203.07404* (2022).
>
> [2] Rudin, Leonid I., Stanley Osher, and Emad Fatemi. "Nonlinear total variation based noise removal algorithms." *Physica D: nonlinear phenomena* 60.1-4 (1992): 259-268.
>
> [3] Chartrand, Rick. "Numerical differentiation of noisy, nonsmooth data." *International Scholarly Research Notices* 2011 (2011)
>
> [4] Brunton, Steven L., Joshua L. Proctor, and J. Nathan Kutz. "Discovering governing equations from data by sparse identification of nonlinear dynamical systems." *Proceedings of the national academy of sciences* 113.15 (2016): 3932-3937.
>
> [5] Vivek S Borkar. Stochastic approximation with two time scales. Systems & Control Letters, 29(5): 291–294, 1997.
>
> [6] Tianyi Chen, Yuejiao Sun, and Wotao Yin. Closing the gap: Tighter analysis of alternating stochastic gradient methods for bilevel problems. Advances in Neural Information Processing Systems, 34: 25294–25307, 2021.
>
> [7] Georg Martius and Christoph H. Lampert. Extrapolation and learning equations.arXiv:1610.02995 [cs], October 2016.

---

> > ### Comment · Reviewer_t2KB · 2022-11-24
> > **After rebuttal: increase my score**
> >
> > I would like to thank the authors for answering precisely and comprehensively to my concerns, and also to other reviewers' questions. I am impressed by the amount of work and additional experiments conducted during this period. I feel now more confident on the principles of this model, and I would like to increase my score.

---

> ### Author Response · Authors · 2022-11-19
> **Response to Reviewer t2KB (Part 2/3)**
>
> **Q5:** The loss minimized in practice is a MSE between the derivatives along the trajectories. How is the derivative estimated for the ground-truth trajectory (finite differences)? Are these estimates noisy with noisy data and how does it affect performances?
>
> **A5:** We use Total Variation Regularization [2,3] for estimating derivatives from noisy data that regularizes the total variation of the estimated derivatives to suppress noise. This has shown effective robustness to noise in prior works [4]. We have added this detail to Section 5. Also, at reviewer’s suggestion, we have added experiments with increasing amounts of noises (1%, 5%, 10% Gaussian noises) in Appendix C.4. We present these results in Figure 7 and observe that MetaPhysiCa is relatively robust to $\leq$ 5% Gaussian noise and significantly outperforms the baselines. With a larger amount of noise, MetaPhysiCa is unable to identify the dynamical system accurately but still performs comparable to the baselines.
>
> **Q6:** In Eq 3, the authors write the global optimization objective. Is this the real bi-level objective optimized in practice?
>
> **A6:** Exact bi-level optimization is challenging to optimize in practice and is typically approximated by alternate SGD steps for inner and outer optimization respectively [5, 6]. In our experiments, joint optimization objective (below) resulted in comparable performance with considerable computational benefits over alternating SGD.
>
> $$\hat{\Phi}, \hat{\xi}, \hat{W}^{(1)}, \ldots, \hat{W}^{(M)}= \text{argmin}\_{\Phi, \xi, {W}^{(1)}, \ldots, {W}^{(M)}} \frac{1}{M} \sum_{i=1}^M R^{(i)}({W}^{(i)}, \Phi, \xi) + \lambda\_{\Phi} ||\Phi||\_1 + \lambda\_{W} \sum_{i=1}^M ||W^{(i)}||\_1 + \lambda\_{\text{REx}} \text{Variance}(\{R^{(i)}({W}^{(i)}, \Phi, \xi)\}\_{i=1}^M) $$
>
> We have added additional implementation details of the proposed method in Appendix B.1 including the joint optimization objective (Equation (6)), basis functions used, and all the hyperparameters.
>
>
>
> **Q7:** What is the link with other recent approaches leveraging causality in PIML, eg [1] ?
>
> **A7:** Thank you for pointing us to this interesting paper on Causal PINNs [1]. There are key differences from our work, especially in the notion of causality. Causal PINNs assume a known physics model (ODE or PDE) and aim to solve this differential equation with the help of NNs. The principle of causality used in their work ensures that, for any time $t$, predictions at time less than $t$ are accurately resolved before predictions at time $t$. Practically, this is implemented using a weighted loss function with weights decreasing over time.
>
> Our work is different in that our notion of causality considers how the relevant system variables affect other variables via a structural causal model, without apriori knowledge of the complete ODE/PDE. Unlike Causal PINNs, this allows for causal reasoning over these variables via interventions. Under our setting, Causal PINNs will suffer from the same problems as other transductive models described in Section 3.2; however, their approach on weighting the loss terms can be employed with MetaPhysiCa. We have added these differences to our related work section.
>
>
>
> **Q8:** The implementation details are missing (in paper or appendix). A schematic of the model could be fine for understanding but is not mandatory. The neural network details should also be reported for the baselines for a fair comparison.
>
> **A8:** Thank you. We have now added additional implementation details of the proposed method and the baselines in Appendix B, along with the respective hyperparameters. We have also added a schematic diagram of MetaPhysica in Figure 6 in the Appendix depicting the meta model $\Phi$, task-specific parameters $W^{(i)*}$, and the corresponding training and test procedures.

---

> ### Author Response · Authors · 2022-11-19
> **Response to Reviewer t2KB (Part 1/3)**
>
> We thank the reviewer for the positive comments and insightful feedback
>
> **Q1:** Could you provide (possibly in appendix) a more thorough analysis? Have you analyzed the discovered causal structure and what interpretation can be drawn?
>
> **A1:** Thank you for your suggestion. We have now provided more qualitative analysis in Appendix C.1 & Table 2 where we show that MetaPhysiCa learns the ground truth ODE (possibly, reparameterized) for all three dynamical systems.
>
>
>
> **Q2:** Why not addressing in the paper the robustness to ODE system parameters?
>
> **A2:** Thank you for the great feedback. We have added an experiment in Appendix C.3 for the damped pendulum system with OOD dynamical system parameters and OOD initial conditions. We briefly describe the experiment and results below.
>
> Consider the damped pendulum system: $\frac{d\theta_t}{dt} = \omega\_t, \frac{d\omega_t}{dt} = -\alpha^{\*2} \sin(\theta_t) - \rho^* \omega\_t$ where $W^*=(\alpha^*, \rho^*)$ are the dynamical system parameters. In training, we sample the dynamical system parameters $\alpha^{(i)\*} \sim \mathcal{U}(1, 2)$ and $\rho^{(i)\*} \sim \mathcal{U}(0.2, 0.4)$, whereas in OOD test, we sample the dynamical system parameters $\alpha^{(i)\*} \sim \mathcal{U}(2, 3)$ and $\rho^{(i)\*} \sim \mathcal{U}(0.1, 0.2)$. Rest of the experimental methodology is kept same as before. Table 2 shows that MetaPhysiCa is able to adapt its parameters to the OOD dynamical system parameters and outputs $\approx 5\times$ more robust OOD predictions compared to the baselines. We will add these experiments to the main paper to provide the reader with a complete picture of all the out-of-distribution scenarios.
>
>
>
> **Q3:** Why not evaluate your method on more complex PDE equations? Published methods evaluate on 2D reaction-diffusion and Navier Stockes (for CoDA) and turbulent flows and sea surface temperature (for DyAd).
>
> **A3:** We are indeed interested in the extension of MetaPhysiCa to more challenging PDE tasks. However, we believe that forecasting PDEs under OOD initial conditions is a separate problem that requires solving several other challenges such as incorporating an expanded set of basis functions that includes differential operators (like the Laplace operator), and considering out-of-distribution boundary conditions (not studied extensively in the literature). While published NN-based methods like CoDA and DyAd are able to forecast PDEs, they are not robust to OOD initial conditions even in simpler cases, as shown in our work. We believe solutions to these challenges will take the focus away from the key ideas of this work, which is to combine meta learning and causal structure discovery to learn OOD-robust models.
>
> We have added the above discussion as a potential avenue for future work in our Conclusions section.
>
>
>
> **Q4:** The experimental section contains only forecasting performances in and out of distribution. Without an in-depth analysis of the model, for example an ablation study, it is hard to understand what elements make this model more successful.
>
>  **A4:** Thanks for the suggestion. We now present an ablation study comparing different components of MetaPhysica in Appendix C.2. Table 3 shows out-of-distribution test NRMSE for the method without each individual component on the three dynamical systems (varying $W^{(i)*}$ scenario). We observe that sparsity regularization (i.e., $||\Phi||_1$) and test-time adaptation are the most important components. For two out of three experiments, the method without sparsity regularization returns errors during prediction. When evaluating MetaPhysiCa without test-time adaptation, we simply use the mean of the task-specific weights learnt for training tasks as the task-specific weight for the given test trajectory, i.e., $\hat{W}^{M+1} = \frac{1}{M} \sum_i \hat{W}^{(i)}$. This results in high OOD errors showing the importance of test-time adaptation. The other two components of the method, i.e., the task-specific $\ell_1$-regularization (i.e., $||W^{(i)}||_1$) and the V-REx penalty help in some experiments and perform comparably in others.

---

> ### Author Response · Authors · 2022-12-09
> **Clarifications to remaining concerns**
>
> We thank the reviewer for engaging with our work. We believe it is an important work to the community and we want to make sure that the paper is well-understood by the community. We are happy to address any unclear points.
>
>
>
> **Q:** The positioning with respect to Coda and DyAd is not discussed and the reason that this method outperform the other by an order of magnitude is still unclear for me.
>
> **A:** In our related work section, we classify CoDA and DyAd under neural network based methods (Section 3.1), as the their out-of-distribution failure is due to their use of neural networks with **no algorithmic alignment** (e.g., with a set of basis functions). CoDA and DyAd are not designed to be robust to out-of-distribution initial conditions and fail OOD for the same reason as NeuralODE (as shown in Figure 1c).
>
> MetaPhysiCa on the other hand is able to learn the true structure of the dynamical system using the meta model $\Phi$ over the given set of basis functions (as shown in Table 2). Having learnt the true structure of the dynamical system, it becomes easier to extrapolate to out-of-distribution initial conditions with an additional test-time adaptation step to learn the task-specific parameters $W^{(i)*}$.
>
> We will expand our results section (Section 5) to emphasize the reason for failure of CoDA and DyAd.
>
>
>
>
>
> **Q:** Besides, I do not understand in experiments why Aphynity is largely inferior to the data-driven Neural ODE for the in-distribution pendulum.
>
> **A:** APHYNITY uses a recurrent encoder to predict the parameters of the given physics model using initial observations $X_{t_0}, …, X_{t_r}$ (as done in Appendix G of APHYNITY [1]). This is required because $W^{(i)\*}$ may vary for every trajectory and we do not assume to have the oracle knowledge of whether $W^{(i)*}$ is constant or varying. In our experiments, we observe that physics parameter prediction using a neural network is harder to train compared to a purely data-driven NeuralODE.
>
> In [1], authors consider a small-data regime, due to which NeuralODE performs worse than APHYNITY. In our experiments, we are not in the small-data regime and a data-driven model is able to fit the in-distribution data almost perfectly.
>
>
>
> **References**
>
> [1] Yuan Yin, Le Vincent, DONA Jérémie, Emmanuel de Bezenac, Ibrahim Ayed, Nicolas Thome, et al. Augmenting physical models with deep networks for complex dynamics forecasting. In International Conference on Learning Representations, 2021.

---

### Official Review · Reviewer_rgk4 · 2022-11-03

**Confidence:** 4
**Correctness:** 2
**Technical Novelty And Significance:** 2
**Empirical Novelty And Significance:** 2
**Recommendation:** 6

**Clarity, Quality, Novelty And Reproducibility:**

The paper is very poorly written, and novelty feels limited. MetaPhysiCa is very close to SINDy but with an added adaptation phase using fine-tuning on test data which is very problematic. Reproducibility is not straightforward as many details are missing.

**Strength And Weaknesses:**

First, I would like to point out that the paper is, in my opinion, very poorly written. Almost all of the arguments are detailed in a convoluted way that is difficult to read, especially section 2 and 3. Figure 1 needs to be re-worked. It is difficult to separate contributions from their poor presentation, so my review may contain misinterpretations.

Many points bother me about the method used. My main concern is on the adaptation phase: it seems that you train your model on the test data. This is a very dangerous choice, which requires a lot of carefulness about how to achieve it. Here are some leads:
- The amount of data on which MetaPhysiCa is adapted in test-time is not so negligible, according to the appendix. The risk of overfitting is huge. I suggest using a validation set from the same distribution as the test set but different to adapt the model, and then test it on a completely unseen set of initial conditions.
- It would also be interesting to see how the model reacts if it is again exposed to the training trajectories after adaptation. My fear is that MetaPhysiCa performs poorly, due to overfitting on testing data.
- When MetaPhysiCa is trained on constant $W*$ tasks, the adaptation phase should be unnecessary. Its ablation in this configuration would confirm that the model is indeed able to identify the causal model.

I strongly recommend that authors demonstrate indisputably that adaptation during test-time is reasonable and relevant.

I also disagree with several claims:
- I see many similarities between MetaPhysiCa and SINDy: the two models seek to identify an analytical model describing the dynamic system. I would have appreciated a much clearer discussion of the differences between MetaPhysiCa and this baseline. I particularly disagree with the sentence "These transductive methods, however, do not transfer knowledge learned in training to predicting test examples unseen during training". SINDy (and EQL to a lesser extent) makes it possible to identify an analytical dynamic equation from the data. This equation is (theoretically) general and therefore perfectly transferable to other initial conditions. The failure of SINDy seems very surprising to me.
- Moreover, MetaPhysiCa seems to me to be able to (even forced to) identify the real dynamic equation. This is a simple check to perform. It would then interesting to verify that the causal graph is correctly identified, which would make it possible to generalize OOD. That being said, I don't see what prevents SINDy from doing the same, especially in the case where the $W*$ parameters are fixed. I would appreciate an in-depth analysis of this.
- The authors justify the failure of methods like APHINITY or NODE by arguing that the neural networks are not algorithmically aligned to the problem. This is quite reasonable, but in my opinion makes the comparison unfair, since the basis functions necessary to solve the problem are directly implemented in MetaPhysiCa by hand. What happens if the structure of NODE is adapted to suit the problem?
- The authors propose constrain their causal graph to be minimal. Although this is a common assumption for the identification of the equations of a dynamical system, a discussion on the relevance of this choice (and its interest in practice) seems to me necessary.

Finally, section 5 shows results that I don't understand:
- The total failure of SINDy and EQL seems very surprising to me. These methods identify an analytical equation which should therefore produce relatively correct results. Their failure requires further analysis.
- The figures presented in addition to the tables show predictions from baselines and MetaPhysiCa. However, I don't understand how the initial conditions (ie the starting point of the curves) can be different from one baseline to another. Could it be that the baselines are evaluated on different trajectories? Authors must justify this.
- I suspect the Deep learning baseline to overfit on the relatively simple system chosen by the authors. However, I could not find information about the size of the networks used for NODE and APHINITY. I believe that the OOD failure of this model maybe tempered with smaller models less subject to overfitting. I would appreciate if the authors could discuss this.

**Summary Of The Paper:**

The paper focuses on dynamical system prediction in the case where the initial conditions of test trajectories are sampled differently from that of training. Metaphysica is introduced, a model based on the identification of causal models which are then finetuned on the new test trajectories to maintain good performance on the new OOD trajectories.

**Summary Of The Review:**

The paper suffers from its poor writing, and its lack of experience and analysis to clearly justify the successes of MetaPhysiCa and the failures of other models. As it stands, I do not recommend this paper for acceptance. Nevertheless, the proposed method is interesting, as is the task, which seems to me to have potential.

## EDIT
The authors provided very convincing answers to my questions. In my opinion, there are still two weaknesses to the proposed method:
1) This is based on a dictionary of functions, and the model assumes that this contains the functions necessary to model the dynamics. This is a difficulty that is also found in SINDy, and many other methods.
2) Experiments are performed on relatively simple dynamical systems. The contribution would have been reinforced if it could have been applied to more complex systems, even real ones, although I am aware of the difficulty of bringing together a dataset for this task.

---

> ### Author Response · Authors · 2022-11-18
> **Response to Reviewer rgk4 (Part 4/4)**
>
>
> **Q13:** Novelty feels limited. MetaPhysiCa is very close to SINDy but with an added adaptation phase using fine-tuning on test data which is very problematic. Reproducibility is not straightforward as many details are missing.
>
> **A13:** Novelty: A major aspect that differentiates MetaPhysiCa from SINDy is the meta-learning framework (with a meta-model $\Phi$ and task-specific parameters $W^{(i)*}$) that allows us to learn from training data consisting of trajectories from different ODE system parameters. Furthermore, to the best of our knowledge, there are no other works that consider the out-of-distribution forecasting task in the generality considered in our work.
>
> Soundness: We hope that we have clarified above that our test-time adaptation that acts only on available test observations for a single test trajectory is sound.
>
> Reproducibility: We have added all the details in Appendix B.1 including the approximate optimization objective, the basis functions used, and the hyperparameters. We will provide anonymized code for better reproducibility.
>
>
>
> **References:**
>
> [1] Wang, Rui, Robin Walters, and Rose Yu. "Meta-learning dynamics forecasting using task inference." *arXiv preprint arXiv:2102.10271* (2021).
>
> [2] Mehta, Viraj, et al. "Neural dynamical systems: Balancing structure and flexibility in physical prediction." *2021 60th IEEE Conference on Decision and Control (CDC)*. IEEE, 2021.
>
> [3] Chen, Ricky TQ, et al. "Neural ordinary differential equations." *Advances in neural information processing systems* 31 (2018).
>
> [4] Yin, Yuan, et al. "Augmenting Physical Models with Deep Networks for Complex Dynamics Forecasting." *International Conference on Learning Representations*. 2021.

---

> > ### Comment · Reviewer_rgk4 · 2022-11-22
> > **Answer**
> >
> > I thank the authors for their complete and convincing answers. There was indeed a misunderstanding that the authors have well justified. I respond in detail to the first answers, since the answers to Q8 to Q12 are convincing and speak for themselves.
> >
> > Q1-Q4: You interpret the adaptation of the model on the first instants of the trajectory as the warm-up phase found on recurrent models, except that MetaPhysiCa adapts in a transductive way. This is actually much less problematic than what my first review suggested. The risk of overfitting is also lower, since the causal structure remains fixed. I really appreciate the new appendix C.1 which, in my opinion, reinforces the message of the paper.
> >
> > Q5: Would it be possible (and useful) to pre-train SINDy and EQL on the training trajectories (although the parameters vary), then fine-tune the models on the test trajectories? Even if SINDy and EQL are not able to generalize to different ODE parameters, it seems to me possible that some information can still be exploited (in particular the null terms). I don't expect these models to outperform MetaPhysiCa, but that might make the comparison fairer.
> >
> > Q6: Thank you for this additional experience.
> >
> > Q7: I was thinking of providing for example additional inputs to NODE and APHINITY, like $x_i^2,\sin(x_i), ...$. I'm not so sure that these models are not able to adapt to new parameters from the measurements $t_0, ..., t_r$, but certainly less easily than MetaPhysiCa.
> >
> > I take note of the effort to improve the paper, the addition of new experiences and the clarification of certain passages. I therefore correct my initial score and lean towards a weak acceptance of the paper.

---

> > > ### Author Response · Authors · 2022-12-03
> > > **Further Response**
> > >
> > > Thank you for taking the time to engage with us to improve the paper. Below we address your remaining concerns.
> > >
> > >
> > > **Q14:** This is based on a dictionary of functions, and the model assumes that this contains the functions necessary to model the dynamics. Experiments are performed on relatively simple dynamical systems … although I am aware of the difficulty of bringing together a dataset for this task.
> > >
> > > **A14:** Our structure causal model can be made more expressive by considering composition of the given basis functions, thereby allowing for more complex dynamical systems. In our new draft (after the original rebuttal phase), we performed an experiment with a 2-layer basis function composition procedure (details in Appendix C.5) on a complex ODE task (adapted from [1]). We describe the experiment and results below briefly for your convenience.
> > >
> > >
> > > Consider for example, 2 given basis functions $f_1(x_t; \xi_1) = \sin(\xi_{1,1} x_{t,1} + \xi_{1,2})$, and $f_2(x_t; \xi_2) = x_{t,1} x_{t,2}$. We construct more expressive basis functions via compositions: $\tilde{f}\_3(x_t;\xi_3)=\sin(\xi_{3,3} \sin(\xi_{3, 1} x_{t, 1} + \xi_{3,2}) + \xi_{3,4})$, $\tilde{f}\_4(x_t;\xi_4)= x_{t,1} x_{t, 2} \sin(\xi_{4,1} x_{t, 1} + \xi_{4,2})$, etc., where $\xi_j$ are global parameters that remain constant for all training/test tasks. The rest of the SCM (Figure 3) remains the same and the derivative $\frac{d{x}^{(i)}_{t, j}}{dt}$ for a particular dimension $j\in \{1,\ldots,d\}$ is a sparse linear combination of the original basis functions and the more expressive second layer ones.
> > >
> > > **Experiment:** We consider a two-dimensional ODE with state $x_t = [p_t, q_t]$: $\frac{dp_t}{dt} = a^* \sin(p_t) + b^* \sin(q_t^2); \frac{dp_t}{dt} = c^* \sin(p_t)\cos(q_t)$, where $W^*=(a^*, b^*, c^*)$ are the dynamical system parameters. In training, we sample initial states $p_t, q_t \sim \mathcal{U}(0.5, 1)$, whereas in out-of-distribution test, we sample $p_t, q_t \sim \mathcal{U}(1, 1.5)$.
> > >
> > > Table 5 shows the results for this task. First, we note that, due to the complexity of a 2-layer learnable basis function procedure, we sometimes need to use validation data (held out from training) to cross-validate the learned model (and reject models that do not do well in validation). In these experiments MetaPhysiCa performs $1.5\times$ to $1.7\times$ better than the competing baselines.
> > >
> > > We believe there is room for improvement in the optimization procedure of these more complex models. **Our updated conclusions section discusses these limitations and possible avenues for future work.**
> > >
> > >
> > >
> > >
> > >
> > >
> > >
> > > **Q15:** Would it be possible (and useful) to pre-train SINDy and EQL (to obtain null terms) on the training trajectories (although the parameters vary), then fine-tune the models on the test trajectories?
> > >
> > > That is an interesting ablation that can be seen as a heuristic approach to find the meta model $\Phi$ (denoting the null terms). MetaPhysiCa uses a more sound approach which leads to a better meta model $\Phi$. We have now performed this ablation experiment. The following results for damped pendulum experiment with varying parameters is in the table below. SINDy with the meta learning heuristic performs better than before (as expected), but still worse than MetaPhysiCa (as we expected because MetaPhysiCa is a more sound approach to finding $\Phi$). EQL even with the heuristic suffers the same issues as before, unfortunately.
> > >
> > > |   | Test Normalized RMSE $\downarrow$ | |
> > > |---------|----------------|---------|
> > > |  **Method** | **ID** |  **OOD** |
> > > | SINDy + Meta Learning heuristic | 0.088 (0.004)   | 0.139 (0.018)
> > > | EQL + Meta Learning heuristic | NaN | NaN |
> > > | MetaPhysiCa | 0.049 (0.002) | 0.070 (0.011) |
> > >
> > > We will add this ablation to the paper.
> > >
> > > **Q16:** I was thinking of providing additional inputs to NODE and APHINITY like xi2, sin⁡(xi). I'm not so sure that these models are not able to adapt to new parameters but certainly less easily than MetaPhysiCa.
> > >
> > > **A16:** NODE and APHYNITY with their neural network terms are prone to learning shortcuts from input to output instead of finding relations related to a causal graph. Adding these basis functions to the input of neural network will make the model larger and is more likely to learn these spurious shortcuts, especially to try to adapt to varying parameters.
> > >
> > > More generally, it is unclear if these models can automatically learn the basis functions in a way that is robust OOD (as shown in Figure 1c), but is not known to be provably impossible yet. From limited support in the training data, can we learn the basis functions such that they correctly extrapolate outside the training distribution support without some mechanistic assumption on how the function extrapolates outside the training support? This question is key to removing dictionary of basis functions.
> > >
> > > **References**
> > >
> > > [1] Gang Chen. Learning symbolic expressions via gumbel-max equation learner networks. arXiv preprint arXiv:2012.06921, 2020

---

> ### Author Response · Authors · 2022-11-18
> **Response to Reviewer rgk4 (Part 3/4)**
>
> **Q9:** The total failure of SINDy and EQL seems very surprising to me. These methods identify an analytical equation which should therefore produce relatively correct results. Their failure requires further analysis.
>
> **A9:** SINDy and EQL’s failure is due to their inability to transfer knowledge from the training data to the test data (MetaPhysiCa uses the training data to learn a meta-model $\Phi$). They can only fit to each separate test trajectory observed from $t_0, …, t _r$, as shown in Figure 1(d), and forecast the future. They are unable to identify an accurate and sparse analytical equation from this small amount of data. MetaPhysiCa, due to the proposed meta-learning framework, is able to learn a meta-model $\Phi$ (an analytical equation independent of task-specific parameters) from the training trajectories and adapt its task-specific parameters for the test trajectory.
>
>
>
> **Q10:** The figures presented in addition to the tables show predictions from baselines and MetaPhysiCa. However, I don't understand how the initial conditions (ie the starting point of the curves) can be different from one baseline to another. Could it be that the baselines are evaluated on different trajectories? Authors must justify this.
>
> **A10:** All the baselines are evaluated on the same ground truth test trajectory (shown as blue stars). All methods are given observations from a test trajectory from time $t_0, …, t_r$ and asked to predict the entire curve from $t_0, …, t_T$. While the results in Tables are computed over time $t_{r+1}, …, t_T$ alone, we show predictions of the entire curve for completeness.
>
> SINDy and EQL are guaranteed to begin predictions from the same initial conditions as given in the input. However, neural network based baselines such as NeuralODE, APHYNITY, DyAd use an encoder to encode the input observations at test $X_{t_0}, …, X_{t_r}$ and use the encoded representation to predict for times $t_0, …, t_T$. Thus, their predicted initial state is not guaranteed to be the same as the input initial state, especially in the OOD regime.
>
>
>
> **Q11:** I suspect the Deep learning baseline to overfit on the relatively simple system chosen by the authors. However, I could not find information about the size of the networks used for NODE and APHINITY. I believe that the OOD failure of this model maybe tempered with smaller models less subject to overfitting. I would appreciate if the authors could discuss this.
>
> **A11:** We have added the architecture sizes and other hyperparameters used for all the baselines in Section B (B.2 for NeuralODE and B.4 for Aphynity) in the Appendix. Our range of hyperparameters contains small neural architectures also, e.g., 1 hidden layer and 32 hidden units as an example, with the hyperparameters chosen based on held-out in-distribution validation data. Thus, the OOD failure of these models is due more to the absence of algorithmic alignment than model size.
>
>
>
> **Q12:** Almost all of the arguments are detailed in a convoluted way that is difficult to read, especially section 2 and 3. Figure 1 needs to be re-worked. It is difficult to separate contributions from their poor presentation, so my review may contain misinterpretations.
>
> **A12:** We believe the difficulty came from a small initial misunderstanding. We have improved the presentation in the updated draft. We have added clarifications in Sections 2-4 emphasizing (i) key aspects of our forecasting task below Definition 1, (ii) differences between existing transductive approaches like SINDy and the proposed approach in Section 3.2, and (iii) more description of the test-time adaptation in Section 4.3. We have also added a schematic diagram of MetaPhysiCa in Figure 6 in the Appendix depicting the meta model $\Phi$, task-specific parameters $W^{(i)*}$, and the corresponding training and test procedures.
>
> We hope we have clarified and improved the presentation. We are happy to discuss more with the reviewer and incorporate any specific improvements to the presentation.

---

> ### Author Response · Authors · 2022-11-18
> **Response to Reviewer rgk4 (Part 2/4)**
>
> **Q4:** When MetaPhysiCa is trained on constant $W^*$ tasks, the adaptation phase should be unnecessary. Its ablation in this configuration would confirm that the model is indeed able to identify the causal model.
>
> **A4:** It is important to note that we **do not** assume to have the oracle knowledge whether $W^{(i)*}$ is constant or not. In our revised draft we now provide a qualitative analysis of the learnt causal structure in Appendix C.1 & Table 2 where we show that MetaPhysiCa learns the ground truth ODE (possibly, reparameterized) for all three dynamical systems.
>
>
>
> **Q5:** I see many similarities between MetaPhysiCa and SINDy: the two models seek to identify an analytical model describing the dynamic system …. I particularly disagree with the sentence "These transductive methods, however, do not transfer knowledge learned in training to predicting test examples unseen during training". The failure of SINDy (and EQL) seems very surprising to me.
>
> **A5:** Unlike MetaPhysiCa, SINDy requires that all training and test trajectories have the same ground truth task-specific parameters $W^{(i)\*}$.
> However, our task definition is more general: each training trajectory $i$ can have different ODE parameter $W^{(i)*}$. Thus, SINDy (and EQL) are unable to use the training data at all when making test predictions. This is illustrated in Figure 1(d) where these models can be fit only on the available observations of each specific test trajectory from time $t_0,...,t_r$ and forecast the trajectory transductively. The sentence "These transductive methods, however, do not transfer knowledge learned in training to predicting test examples unseen during training" refers to these models’ inability to transfer to different in-distribution ODE system parameters and not the initial conditions. We have clarified this sentence in Section 3.2.
>
>
>
> **Q6:** Moreover, MetaPhysiCa seems to me to be able to (even forced to) identify the real dynamic equation. This is a simple check to perform. That being said, I don't see what prevents SINDy from doing the same.
>
> **A6:** Thank you for this suggestion. In the updated draft have now provided a qualitative analysis of the learnt causal structure in Appendix C.1 & Table 2 where we show that MetaPhysiCa learns the ground truth ODE (possibly, reparameterized) for all three dynamical systems.
>
> Failure of SINDy: Our task definition is general where each training trajectory $i$ can have different ODE parameter $W^{(i)*}$. Further, in our experiments, we always consider the harder and realistic scenario with no **oracle knowledge** of which scenario (constant or varying $W^{(i)\*}$) the data is observed from. Without such oracle knowledge, SINDy is unable to use the training data at all and fits only on the available observations of the specific test trajectory at time $t_0,...,t_r$ and forecasts each trajectory transductively. Thus, SINDy is unable to identify an accurate and sparse analytical equation from this small amount of data. We have added these clarifications to our Empirical Evaluation section (Section 5).
>
>
>
> **Q7:** The authors justify the failure of methods like APHINITY or NODE by arguing that the neural networks are not algorithmically aligned to the problem. What happens if the structure of NODE is adapted to suit the problem?
>
> **A7:** Our claim is that these basis functions (algorithmic alignment) are necessary for OOD extrapolation as seen from the failure of standard neural networks in Figure 1(c). We are currently unaware of methods that adapt NODE to include basis functions. One possible way would be to use EQL-type architecture with NODE instead of a feedforward neural network, i.e., $dX_t/dt = \text{EQL}(X_t)$. However, without our proposed meta learning framework, this will suffer the same problems as EQL and be unable to transfer knowledge from training trajectories (with different $W^{(i)*}$ per task) to test trajectory.
>
>
>
> **Q8:** The authors propose to constrain their causal graph to be minimal. A discussion on the relevance of this choice (and its interest in practice) seems to me necessary.
>
> **A8:** We have updated the text with: “We wish to find the minimal causal structure, i.e., with the least number of edges, that also fits the training data. This balances the complexity of the causal structure with training likelihood, and avoids overfitting the training data. A sparse structure for $\Phi$ implies fewer terms in the RHS of the learnt equation for the derivatives in Equation (2).“

---

> ### Author Response · Authors · 2022-11-18
> **Response to Reviewer rgk4 (Part 1/4)**
>
> We thank the reviewer for their time and constructive feedback. We believe the reviewer's concerns are the consequence of a small initial misunderstanding that cascaded throughout the paper. We believe we have clarified this in the updated draft, which we preview here for your convenience:
>
> **Q1:** Many points bother me about the method used. My main concern is on the adaptation phase: it seems that you train your model on the test data. This is a very dangerous choice, which requires a lot of carefulness about how to achieve it.
>
> **A1:** We have added the following clarifications to the paper in Section 4.3. We have also added a schematic diagram of MetaPhysica in Figure 6 in the Appendix depicting the meta model $\Phi$, task-specific parameters $W^{(i)*}$, and the corresponding training and test procedures.
>
> **Test data forecasting task:** We are given a new trajectory (in our paper indexed as M+1) in the window $t = t_0, …, t_r$. We wish to forecast its future after time $t_r$. **This is a common task, see [1,2,3] and [4; Appendix G]**. For instance, NeuralODE uses a recurrent encoder to take $X^{(M+1)}\_{t_0}, …, X^{(M+1)}\_{t_r}$ as input and predict the entire trajectory. Note that, like NeuralODE, we also assume we have $X^{(M+1)}\_{t_0}, \ldots, X^{(M+1)}\_{t_r}$. In standard physics modeling, we would fit the free model physics parameters to $X^{(M+1)}\_{t_0}, \ldots, X^{(M+1)}\_{t_r}$. What we do not observe is $X^{(M+1)}\_{t_{r+1}},\ldots$, which we need to predict. In our OOD test scenario, the initial condition $X^{(M+1)}\_{t_0}$ was never observed in training. The **unknown ODE system parameters $W^{(M+1)*}$** have the same distribution (not necessarily the same values) as the ODE system parameters of the training data.
>
> Since (like in [1,2,3] and [4; Appendix G]) we observe $X^{(M+1)}\_{t_0}, \ldots, X^{(M+1)}\_{t_r}$, we can adapt our model using these initial observations. The adaptation is not "training the model on test data". Prior works [2,3] and [4; Appendix G] perform this adaptation inductively. We perform this adaptation transductively, using a meta-learning procedure to adapt only some parts of the model (the meta model $\Phi$) . Equation (4) in the paper describes our test-time adaptation for a **single** trajectory only over the **observed times** $t_0,...,t_r$. Note the following two key aspects of this adaptation: (i) Only the task-specific parameters $W^{(M+1)}$ are adapted whereas the meta-model $\Phi$ that identifies the dynamical system remains the same, and (ii) Our method does not use the entire test data at once to adapt, but only the available observations of that particular test trajectory $M+1$.
>
> **Training data:** Training trajectories may have (a) different initial conditions $X^{(i)}\_{t_0}$, and **(b) different unknown ODE system parameters $W^{(i)\*}$,** for each trajectory (task) $i$. In our damped pendulum example (described in Page 3 below Definition 1), the training data consists of trajectories from multiple pendulum experiments with different initial dropping angles (initial conditions $X^{(i)}_{t_0}$) and different damping coefficients (different ODE system parameters $W^{(i)*}$). We wish to learn the general dynamical system describing pendulum motion independent of the specific ODE parameters.
>
> We have added emphasis on these key aspects in our paper.
>
>
>
>
>
> **Q2:** The amount of data on which MetaPhysiCa is adapted in test-time is not so negligible, according to the appendix. The risk of overfitting is huge.
>
> **A2:** In our experiments we observe that the risk of overfitting is well-controlled by the meta-model $\Phi$, which remains fixed for the test data. The task-specific parameters for a test trajectory $W^{(M+1)}$ are allowed to vary freely independent of the task-specific parameters $W^{(i)}$ learnt for training tasks $i=1,\ldots,M$. This allows MetaPhysiCa to forecast test trajectories with out-of-distribution ODE parameters $W^{(M+1)*}$ as well as out-of-distribution initial conditions, as shown in Appendix C.3 of our updated draft.
> The amount of test data given to MetaPhysiCa is the same as that available to all other models, i.e., the observations from $t_0, …, t_r$ of a specific test trajectory.
>
>
> **Q3:** It would also be interesting to see how the model reacts if it is again exposed to the training trajectories after adaptation. My fear is that MetaPhysiCa performs poorly, due to overfitting on testing data.
>
> **A3:** There is no such risk. If MetaPhysiCa is shown training trajectories again, it adapts its task-specific parameters to each training trajectory while keeping the learnt meta-model $\Phi$ fixed. Thus, it still forecasts accurately with normalized RMSEs of 0.03, 0.008 and 0.007 for damped pendulum task, predator-prey system and epidemic modeling, respectively.

---

### Official Review · Reviewer_MowG · 2022-11-05

**Confidence:** 2
**Correctness:** 4
**Technical Novelty And Significance:** 4
**Empirical Novelty And Significance:** 4
**Recommendation:** 8

**Clarity, Quality, Novelty And Reproducibility:**

The paper is well-written and easy to follow, very novel, and with sufficient reproducibility.

**Strength And Weaknesses:**

Strength:
I like the motivation of this paper, and the illustrative examples are effectively demonstrating the limitations of vanilla PIML.

Weakness:
I would prefer that the authors state clearly the current limitation of this approach. The SCM adopted in this paper would not scale to the realistic, complex ODE cases, and either some experimental investigation or discussions on the limit of this approach would be beneficial.

**Summary Of The Paper:**

This paper proposed MetaPhysiCa, a treatment to OOD initial conditions in PIML, via meta-learning algorithms with structural causal models.


**Summary Of The Review:**

In summary, this is an interesting paper. I am willing to raise my score if the authors could conduct a more comprehensive investigation on the current scalability of the proposed approach.

---

> ### Author Response · Authors · 2022-11-18
> **Response to Reviewer MowG**
>
> We thank the reviewer for their positive comments and feedback.
>
> **Q1:** I would prefer that the authors state clearly the current limitation of this approach. The SCM adopted in this paper would not scale to the realistic, complex ODE cases, and either some experimental investigation or discussions on the limit of this approach would be beneficial.
>
> **A1:** Thank you for the feedback. In the updated draft we extended MetaPhysiCa to consider significantly more expressive structural causal models that allow for composition of the basis functions. This is achieved with a 2-layer learnable basis function composition procedure (detailed in Appendix C.5). We have also added an experiment on a more complex ODE task (adapted from [1]). We describe the experiment and results below briefly.
>
>
> Consider for example, 2 given basis functions $f_1(x_t; \xi_1) = \sin(\xi_{1,1} x_{t,1} + \xi_{1,2})$, and $f_2(x_t; \xi_2) = x_{t,1} x_{t,2}$. We construct more expressive basis functions via compositions: $\tilde{f}\_3(x_t;\xi_3)=\sin(\xi_{3,3} \sin(\xi_{3, 1} x_{t, 1} + \xi_{3,2}) + \xi_{3,4})$, $\tilde{f}\_4(x_t;\xi_4)= x_{t,1} x_{t, 2} \sin(\xi_{4,1} x_{t, 1} + \xi_{4,2})$, etc., where $\xi_j$ are global parameters that remain constant for all training/test tasks. The rest of the SCM (Figure 3) remains the same and the derivative $\frac{d{x}^{(i)}_{t, j}}{dt}$ for a particular dimension $j\in \{1,\ldots,d\}$ is a sparse linear combination of the original basis functions and the more expressive second layer ones.
>
> **Experiment:** We consider a two-dimensional ODE with state $x_t = [p_t, q_t]$: $\frac{dp_t}{dt} = a^* \sin(p_t) + b^* \sin(q_t^2); \frac{dp_t}{dt} = c^* \sin(p_t)\cos(q_t)$, where $W^*=(a^*, b^*, c^*)$ are the dynamical system parameters. In training, we sample initial states $p_t, q_t \sim \mathcal{U}(0.5, 1)$, whereas in out-of-distribution test, we sample $p_t, q_t \sim \mathcal{U}(1, 1.5)$.
>
> Table 5 shows the results for this task. First, we note that, due to the complexity of a 2-layer learnable basis function procedure, we sometimes need to use validation data (held out from training) to cross-validate the learned model (and reject models that do not do well in validation). In these experiments MetaPhysiCa performs $1.5\times$ to $1.7\times$ better than the competing baselines. We believe there is room for improvement in the optimization procedure of these more complex models.
>
>
> We have expanded our Conclusions section with a discussion on the limitations and possible avenues for future work as follows:
>
> “We believe that forecasting models should be robust to OOD shifts, and that our work takes a step in the right direction with several potential avenues for future research:
>
> **(i)** Partial differential equations (PDEs): Extending MetaPhysiCa to forecasting PDEs under OOD scenarios is an interesting extension that requires an expanded set of basis functions that includes differential operators (like the Laplace operator), and considering out-of-distribution boundary conditions.
>
> **(ii)** More expressive structural causal models (SCMs): Our experiment on a complex ODE task (in Appendix C.5) suggests that MetaPhysiCa with a more expressive SCM that allows for composition of basis functions is able to forecast out-of-distribution better than competing baselines, but suffers from learning stiff ODEs due to the complexity of a 2-layer learnable basis function procedure. Better optimization techniques may help alleviate this problem. “
>
>
> **References:**
>
> [1] Gang Chen. Learning symbolic expressions via gumbel-max equation learner networks. arXiv preprint arXiv:2012.06921, 2020

---

> > ### Comment · Reviewer_MowG · 2022-11-24
> > **After rebuttal**
> >
> >  Thank you for the detailed response. I have increased my score.

---

### Decision · Program_Chairs · 2023-01-20

**Decision:**

Reject

**Justification For Why Not Higher Score:**

- Poor paper presentation
- Lack of positioning with respect to the most related works from the literature
- Consolidations in experiments

**Justification For Why Not Lower Score:**

N/A

**Metareview: Summary, Strengths And Weaknesses:**

This paper deals with the problem of robustness to initial conditions in physics-informed machine learning. The authors point out the brittleness of state-of-the-art methods with respect to changes in initial conditions' distribution, especially when they fall outside the training support. They introduces an approach for improving this robustness, based on a combination of causal structural discovery, meta-learning, and hybrid transductive-inductive test-time adaptation. Experiments are conducted on 3 datasets: Damped pendulum, predator-prey systems, and epidemic modelling. \
The paper initially received mixed reviews, with two borderline reject (5), one reject (3), and two borderline accept (6) recommendations. The main concerns pointed out by reviewers related to the clarity of the presentation, unjustified claims, and several issues in experiments, especially related to the performances of the baselines. The author's feedback was very detailed, with long answers and some new experiments and ablation provided. After rebuttal, the reviewers' grades have been increased to 6 6 6 6 8\
The AC's own reading of the paper reveal additional shortcoming in the paper:
- The claimed contributions related to adapting the model in OOD initial condition setting is based on a combination of meta-learning, causal discovery and hybrid transductive-inductive adaptation. The related work on these topics is, however, not discussed in section 3. One would at least expect a short literature review on these field and how they should be adapted in the context of the submission. Also, there is no positioning with respect to the adaptation methods compared in the experiments, e.g. DyAd and CoDA.
- The paper is poorly presented. Until section 4, the paper mixes up general elements of motivation or experimental results. The method description starting at the end of page 5 is too short, insufficiency illustrated and detailed, making its insights unclear and difficult to follow.
- Experimental conclusions are unclear: the split on methods between deep learning and PIML questionable, since the classification would expected to be between adaptation and non-adaptation methods. The baseline seems not have been properly tuned or evaluated in artificial contexts where they fail. At the end, the proposed approach outperforms recent adaptations method by an order of magnitude, and the explanation of this large gain is left without a thorough analysis.

During the virtual meeting RhnK8, Rrgk4, and Rt2KB discussed about the points raised by the AC. Overall, the weaknesses highlighted by the AC echo some of their own concerns, but added some points that they overlooked. The combination of issues make the reviewers agree that the paper should not be accepted in the current form (see below). RBmNe was not present in the virtual meeting but participated to the discussion: he also hear the AC's point and finally voted for weak rejection.
After the virtual AC meeting, there was a consensus among active reviewers that the submission would highly benefit from a full re-writing and a clarification on experiments.

The AC considers that the paper addresses an interesting and important problem in physics-informed machine learning, and the general idea of combining causal structural discovery, meta-learning, and hybrid transductive-inductive test-time adaptation is promising. However, the submission cannot be accepted in the current state. The proposed method should be expanded, the insights given in a clearer way to reach a broad audience, and the positioning needs to be specified. The experiments should specify the problem statement and justify the choice of the baselines.
The AC thus recommends rejection but highly encourages the authors to re-submit their works based on reviewers' feedback.

**Summary Of Ac-Reviewer Meeting:**


RhnK8, Rrgk4, Rt2KB, RBmNe overlooked the lack of positioning with respect to meta-learning, causal discovery and hybrid transductive-inductive adaptation, but consider that it is an important point. They weaknesses also echo some of their own concerns, e.g.:
- Poor paper presentation (Rrgk4) - Rt2KB and RhnK8 did not see it as a major mistakes essentially because they had worse submissions in their batches
- Limiting assumption that the basis functions necessary to solve the problem are directly implemented in MetaPhysiCa by hand (Rrgk4 and RhnK8)
- Baselines' results in experiments: APHYNITY should not be substantially below deep learning methods in ID settings (rt2KB), or SINDy is unable to converge is an artificial choice of the authors' setup (RhnK8)